# Predicting fracture risk in patients with chronic obstructive pulmonary disease: a UK-based population-based cohort study

Ralph Kwame Akyea,[1,2] Tricia M McKeever,[1,2] Jack Gibson,[2] Jane E Scullion,[3] Charlotte E Bolton[1]

[1]Nottingham Respiratory Research Unit, NIHR Nottingham Biomedical Research Centre, School of Medicine, University of Nottingham, Nottingham, UK
[2]Division of Epidemiology & Public Health, School of Medicine, University of Nottingham, Nottingham, UK
[3]Institute for Lung Health, University Hospitals of Leicester Glenfield Site, Leicester, UK

**Correspondence to**
Professor Charlotte E Bolton; charlotte.bolton@nottingham.ac.uk

## ABSTRACT

**Objective** To assess the incidence of hip fracture and all major osteoporotic fractures (MOF) in patients with chronic obstructive pulmonary disease (COPD) compared with non-COPD patients and to evaluate the use and performance of fracture risk prediction tools in patients with COPD. To assess the prevalence and incidence of osteoporosis.

**Design** Population-based cohort study.

**Setting** UK General Practice health records from The Health Improvement Network database.

**Participants** Patients with an incident COPD diagnosis from 2004 to 2015 and non-COPD patients matched by age, sex and general practice were studied.

**Outcomes** Incidence of fracture (hip alone and all MOF); accuracy of fracture risk prediction tools in COPD; and prevalence and incidence of coded osteoporosis.

**Methods** Cox proportional hazards models were used to assess the incidence rates of osteoporosis, hip fracture and MOF (hip, proximal humerus, forearm and clinical vertebral fractures). The discriminatory accuracies (area under the receiver operating characteristic [ROC] curve) of fracture risk prediction tools (FRAX and QFracture) in COPD were assessed.

**Results** Patients with COPD (n=80 874) were at an increased risk of fracture (both hip alone and all MOF) compared with non-COPD patients (n=308 999), but this was largely mediated through oral corticosteroid use, body mass index and smoking. Retrospectively calculated ROC values for MOF in COPD were as follows: FRAX: 71.4% (95% CI 70.6% to 72.2%), QFracture: 61.4% (95% CI 60.5% to 62.3%) and for hip fracture alone, both 76.1% (95% CI 74.9% to 77.2%). Prevalence of coded osteoporosis was greater for patients (5.7%) compared with non-COPD patients (3.9%), p<0.001. The incidence of osteoporosis was increased in patients with COPD (n=73 084) compared with non-COPD patients (n=264 544) (adjusted hazard ratio, 1.13, 95% CI 1.05 to 1.22).

**Conclusion** Patients with COPD are at an increased risk of fractures and osteoporosis. Despite this, there is no systematic assessment of fracture risk in clinical practice. Fracture risk tools identify those at high risk of fracture in patients with COPD.

### Strengths and limitations of this study

► This study examined electronic health records from a large, nationally representative sample of the UK population.
► A wide range of potential confounders were evaluated and adjusted for in the analyses.
► For the assessment of the fracture prediction tools, the population of patients with COPD used was large, with many fracture (hip alone and all MOF) events, and it included both men and women.
► READ codes recorded in UK primary care do not capture free text from consultations but capture new diagnoses, such as diagnosed osteoporosis, and significant fractures, such as those classed as MOF.
► The incidence of diagnosed osteoporosis based on clinical codes may reflect an underestimation of the true risk of osteoporosis since bone mineral density is not systematically assessed.

## INTRODUCTION

Osteoporosis in both male and female patients with chronic obstructive pulmonary disease (COPD) is firmly established as one of the core comorbid conditions.[1 2] Over the last decade, it has become clear that osteoporosis is not just an end-stage COPD problem[3] nor just in those on maintenance oral corticosteroids (OCS), but it also occurs in a large proportion of those with mild–moderate airflow obstruction and even in steroid naïve patients.[4 5] The Global Initiative for COPD (GOLD) strategy recommends that osteoporosis co-existence should be considered in COPD,[1] and that the UK National Institute for Health and Care Excellence (NICE) Guidelines on osteoporosis considers COPD as a secondary cause of osteoporosis encouraging the use of fracture prediction tools.[6]

The causes for osteoporosis in COPD are likely multiple and cumulative, including age, smoking exposure, inactivity, low body mass index (BMI), systemic inflammation and the

frequent use of OCS.[7] The clinical implications of osteoporosis include increased risk of fractures, poor quality of life, pain and further deterioration in lung function.[8 9] Osteoporosis can also remain undiagnosed as asymptomatic for many years.[10] Fractures are a function of trauma sustained, such as falls which are common in COPD,[11] and affect the quality and architecture of the bone. Fractures contribute further pain, poor quality of life, increased mortality and confer a substantial economic burden on health systems, patients and their families.[12 13] Given this, the individual risk of a future fracture in patients with COPD is crucial to determine in-patient care and to treat accordingly.

Fracture risk prediction tools based on clinical and personal characteristics have been developed over the years to guide the investigation and management of those identified to be at high risk of osteoporotic fractures, worldwide. These include for the UK (and many other regions), FRAX and QFracture.[6]

The full extent of fracture risk assessment in patients with COPD is not fully established. The aim of this study was to assess the incidence of hip fracture alone or all major osteoporotic fractures (MOF) in patients with COPD compared with non-COPD patients, and to evaluate the use and performance of fracture risk prediction tools in patients. Further, to assess the prevalence of coded osteoporosis up to the time of COPD diagnosis and the incidence of osteoporosis.

## METHODS

Information for this cohort study was obtained from The Health Improvement Network (THIN), an anonymised primary care database representing 6.2% of the total UK population.[14]

The study population consisted of patients 40 years and over with a new READ-coded COPD diagnosis during the data collection period 1st January 2004 to 31st December 2015, with at least 1 year of record prior to COPD diagnosis.[15] Each patient was matched by age, sex and general practice (GP) to up to four patients without a history of COPD to generate a matched cohort and assigned the same index date.

Follow-up was from the index date to the first record of either the occurrence of the outcome of interest (fracture/osteoporosis), the date of transfer of the patient out of the practice area, death or the end of THIN data collection. Diagnoses for osteoporosis—classed as coded osteoporosis (online appendix 1), hip fracture alone and all MOF (comprising fracture of the hip, proximal humerus, forearm or clinically symptomatic vertebra/spine), coded using the standard READ code classification were used.[16]

A series of explanatory variables[6 17] determined at baseline (prior to or at index date) included Charlson Comorbidity Index (CCI) score,[18] Townsend social deprivation score, recorded history of fall, prior fractures, parental history of fall/osteoporosis, relevant comorbidities and secondary causes of osteoporosis as defined in the FRAX

questionnaire.[19] Records for smoking status, alcohol use, Medical Research Council (MRC) Dyspnoea scale, BMI and use of specific prescription drugs were restricted to a defined time period.

To account for the use of OCS, individual follow-up time was divided into periods during which participants were considered exposed, or not exposed, to OCS (a binary variable). Exposed periods started from prescription date until the first gap of more than 90 days between prescriptions, with individuals considered unexposed from the 91st day onwards. Individuals were considered exposed at study entry if they had received a relevant prescription within 90 days prior. The effect of exposure was assumed to be constant, and not cumulative, over time.

Input variables included clinical status, prescription drug use, and demographic characteristics, according to the variables/definitions used in both FRAX and QFracture tools,[19 20] additional detail on the method is provided in an online data supplement (online appendix 2). Imputation was used for missing variables.

The 10-year risk score for hip fracture alone and all MOF according to QFracture (V.2017.0.0.0) and FRAX for UK (without bone mineral density [BMD] information) (desktop V.3.12) were calculated for patients with COPD, aged 40–90 years. A complete case sensitivity analysis without imputed variables was also performed (online appendix 3).

### Statistical analyses

Incidence rates were calculated for both groups using Cox proportional hazards regression to estimate HRs of coded osteoporosis and fracture (hip alone and all MOF) risks. Conditional analysis to account for matching by age, sex and GP practice was done. Confounders were included in the final fully adjusted multivariable models when independently changing the HRs for osteoporosis/fracture by at least 5%. A former osteoporosis diagnosis or antiresorptive treatment prior to COPD diagnosis excluded that subject from analyses related to either osteoporosis incidence or risk (online appendix 4). In addition to evaluating incidence in the whole cohort, separate sub-analyses excluded (a) patients with COPD and no documented smoking history together with their matched non-COPD patients and (b) those with no prior record of osteoporosis.

To evaluate FRAX and QFracture, the outcome was treated as a binary variable (fracture or no fracture). Fracture risk probabilities were categorised based on recommended treatment thresholds (≥20% for MOF and ≥3% for hip fracture).[21] To evaluate the overall ability of each tool to discriminate (performance) between those at low and high risks, the area under the receiver operating characteristic (ROC) curve was calculated. Sensitivity, specificity, and positive and negative predictive values were calculated. Survival analysis was performed and Kaplan-Meier plots comparing the MOF incidence were generated.

All statistical analyses were performed using Stata 15.0 (StataCorp LP).

## Patient involvement

The results and implications of previous research from the team on the systematic assessment of osteoporosis in patients with COPD[4] have been discussed extensively in previous patient meetings. While this and other literature have strengthened the GOLD strategy recommendations,[1] evaluation of clinical services would suggest systematic assessment is not done in patients. More recently, patients with COPD out-patient clinics have approached the principal investigator at the time of their 'ad hoc osteoporosis' diagnosis to ask why this was not investigated at or closer to COPD diagnosis and how osteoporosis could be assessed. This has led to the development of this grant application with significant patient input in the design and context. The results have been discussed back with representatives on the respiratory research panel. Given the implications for clinical practice, the findings have been discussed extensively at the patient and public involvement meeting and a Breathe Easy meeting in early 2018. A lay summary has been developed for the patient newsletter (n>700) and website. In the meantime, members of the respiratory research panel are assisting the PI in planning future work regarding implementation.

## RESULTS

The baseline characteristics are shown in table 1. A total of 80874 eligible patients with COPD and 308999 matched non-COPD patients were identified. The median follow-up period was 5 years for both patients with COPD and non-COPD patients.

## Osteoporosis at index date and incidence

The prevalence of coded osteoporosis up to the index date was greater for patients with COPD (5.7%) compared with non-COPD patients (3.9%), p<0.001. Within 1year (before and after) of the index date, 1504 (1.86%) patients with COPD had a new recorded diagnosis of osteoporosis compared with 3059 (1.12%) in matched non-COPD patients, p<0.001. Three thousand one hundred and eighty-six (3.94%) patients with COPD had a diagnosis of osteoporosis more than a year prior to index date compared with 8822 (2.86%) for the matched controls, p<0.001.

One thousand four hundred and fifty-seven (1.80%) patients with COPD compared with 3694 (1.20%) non-COPD patients had a record of any diagnostic assessment for osteoporosis, recorded within 1year (before and after) of the index date, p<0.001.

Demographics remained similar after excluding those with former coded osteoporosis. Patients with COPD (n=73084) compared with non-COPD patients (n=264544) were significantly more likely to have an incident diagnosis of osteoporosis (HR, 1.96; 95% CI 1.87 to 2.05; online appendix 5).

## Incidence of fracture

There was a significantly increased risk of MOF, HR of 1.60 (95% CI 1.52 to 1.69) and hip fractures alone: 1.67 (95% CI 1.56 to 1.80) in patients with COPD compared with non-COPD patients. In the fully adjusted models, the associations were diminished (table 2). Smoking status altered the effect between COPD and fracture the most, followed by BMI, CCI score and OCS.

Sensitivity analysis with participants with no former osteoporosis showed similar results. The risk of MOF was also similar when evaluated in patients only with COPD with a documented prior history of smoking and their matched controls. However, here, the risk of hip fracture remained significantly increased in the adjusted model compared with non-COPD patients (aHR, 1.13; 95% CI 1.004 to 1.280; p-value: 0.043).

## Fracture risk prediction tools in COPD

Only 1074 (1.33%) of the patients with COPD had a FRAX assessment READ-coded ever documented in the records and 12 patients had a READ coded QFracture assessment. Within 1year (before and after) of index date, 248 (0.31%) of the patients with COPD had a FRAX and only one patient a QFracture.

The final population for the discriminatory accuracy analysis comprised 72559 patients aged 40–90 years with COPD and no prior diagnosis of osteoporosis or prescription of any anti-resorptive treatment (demographics in onlineappendix 6). This included 4605 (6.4%) patients who experienced any MOF and 1444 (2.0%) who experienced a hip fracture.

When the FRAX and QFracture scores were calculated for patients with COPD, for hip fracture there were 29035 (40.0%) patients who had a risk≥3% using FRAX and 33065 (45.6%) patients using QFracture. For any MOF, 6221 (8.6%) of the patients had a risk≥20% using FRAX and 9546 (13.2%) patients using QFracture.

Both risk tools had a similar discriminatory accuracy for hip fracture (FRAX 76.1%, 95% CI 74.9% to 77.2% and QFracture 76.1%, 95% CI 74.9% to 77.2%). FRAX, however, had a higher accuracy for MOF (71.4%–95% CI 70.6% to 72.2%) than QFracture (61.4%–95% CI 60.5% to 62.3%). The discriminatory accuracies were better in women than men. The performance of the prediction tools was similar in patients aged 50–90 years compared with those aged 40–90years.

The sensitivity of the risk scores for any MOF (using >20% risk as cut-off) was similar: FRAX: 25.4% and QFracture: 25.2%. The sensitivity of the risk scores for hip fracture (using >3% cut-off) was slightly worse for FRAX: 78.1% compared with 82.1% for QFracture. The specificity and positive predictive value were better for FRAX than QFracture, table 3.

The association of an increased fracture risk (either FRAX or QFracture) with the incidence of any MOF is shown in figure 1.

**Table 1** Baseline characteristics of patients with COPD and non-COPD patients

| Descriptor | COPD patients | | Non-COPD patients | | |
|---|---|---|---|---|---|
| | n=80874 | % | n=308999 | % | P value |
| Mean age at index date (years, SD) | 66.9 (11.0) | | 66.5 (10.9) | | |
| Sex | | | | | 0.002 |
| Male | 42799 | 52.9 | 161648 | 52.3 | |
| Female | 38075 | 47.1 | 147351 | 47.7 | |
| Follow-up (years, median, IQR) | 5.28 | 2.6–8.3 | 5.24 | 2.6–8.3 | |
| **MRC Dyspnoea Scale** (1 year either side of diagnosis) | | | | | <0.001 |
| 1 | 9499 | 11.8 | 1168 | 0.4 | |
| 2 | 19466 | 24.1 | 1092 | 0.4 | |
| 3 | 10488 | 13.0 | 446 | 0.1 | |
| 4 & 5 | 5237 | 6.5 | 177 | 0.1 | |
| No record | 36184 | 44.7 | 306116 | 99.1 | |
| CCI score | | | | | <0.001 |
| 0 | 0 | 0.0 | 172566 | 55.9 | |
| 1 | 41777 | 51.7 | 50955 | 16.5 | |
| 2 | 13506 | 16.7 | 42667 | 13.8 | |
| 3 | 12694 | 15.7 | 23546 | 7.6 | |
| ≥4 | 12897 | 16.0 | 19265 | 6.2 | |
| BMI (kg/m$^2$) | | | | | <0.001 |
| Underweight (<18.5) | 3414 | 4.2 | 2699 | 0.9 | |
| Normal (18.5–24.9) | 24734 | 30.6 | 54267 | 17.6 | |
| Overweight (25–29.9) | 23497 | 29.1 | 77129 | 25.0 | |
| Obese (≥30) | 19083 | 23.6 | 60280 | 19.5 | |
| No BMI | 10146 | 12.6 | 114624 | 37.1 | |
| **Smoking status** (1 year either side of diagnosis) | | | | | <0.001 |
| Never smoked | 7925 | 9.8 | 94800 | 30.7 | |
| Ex-smoker | 38590 | 47.7 | 72989 | 23.6 | |
| Current smoker | 32436 | 40.1 | 34691 | 11.2 | |
| Unknown | 1923 | 2.4 | 106519 | 34.5 | |
| **History of falls** (prior to or at diagnosis) | | | | | |
| Personal history | 8969 | 11.1 | 26203 | 8.5 | <0.001 |
| Parental history of fall/osteoporosis | 96 | 0.1 | 298 | 0.1 | 0.076 |
| **Medications** (1 year either side of diagnosis) | | | | | |
| OCS use | 33618 | 41.6 | 19479 | 6.3 | <0.001 |
| Inhaled corticosteroid use | 47574 | 58.8 | 21312 | 6.9 | <0.001 |

BMI, body mass index; CCI, Charlson Comorbidity Index; IQR, interquartile range; MRC, Medical Research Council; OCS, oral corticosteroids.

## DISCUSSION

Using UK primary care electronic health records, we have reported on the burden of fractures in patients with COPD with both hip fracture alone or any MOF increased in patients with COPD compared with age, sex and GP surgery matched patients. Despite the increased fracture risk and recommendations in the NICE osteoporosis guidelines, fracture risk prediction tools are rarely coded.

However, where the risk score was retrospectively calculated, the risk prediction tools identify those at risk of hip fracture or any MOF. Therefore, fracture risk prediction and subsequent targeted therapy and management could transform multi-morbidity management of COPD. In addition, we report that the prevalence and incidence of osteoporosis, a risk for fracture, in patients with COPD, is far greater than in non-COPD patients.

**Table 2** Risk of all MOF and hip fractures alone in patients with COPD compared with non-COPD patients

| | Number of fractures | Rate/1000 person-years | HR (95% CI) | Fully adjusted HR (95% CI) |
|---|---|---|---|---|
| **MOF** | | | | |
| Non-COPD patients | 6032 | 4.32 (4.22–4.44) | Reference | Reference |
| Patients with COPD | 2234 | 6.64 (6.37–6.92) | 1.60 (1.52 to 1.69) | 1.04 (0.96 to 1.12)* |
| Hip fracture | | | | |
| Non-COPD patients | 3170 | 2.26 (2.18–2.34) | Reference | Reference |
| Patients with COPD | 1213 | 3.57 (3.38–3.78) | 1.67 (1.56 to 1.80) | 1.09 (0.98 to 1.21)† |

Fully adjusted:
*Multivariable Cox regression model-derived HR was adjusted for age, sex, GP, CC I, BMI, smoking status, inhaled corticosteroid use, antidepressant use and cumulative OCS use.
†Multivariable Cox regression model-derived HR was adjusted for age, sex, GP, CCI, BMI, smoking status, inhaled corticosteroid use and cumulative OCS use.
BMI, body mass index; CCI, Charlson Comorbidity Index; COPD, chronic obstructive pulmonary disease; GP, general practice; MOF, major osteoporotic fractures; OCS, oral corticosteroid; HR, conditional regression used to account for matching by age, sex and GP.

Prevalence of osteoporosis varies widely in the different research studies of patients with COPD. This is likely due to the severity of COPD,[4 5] whether osteoporosis was systematically sought or self-reported,[4 22] and whether patients included were on OCS.[3] A prevalence of 23%–32% has been reported where BMD was systematically performed in COPD,[4 23] while 14% of the patients with COPD self-reported osteoporosis compared with 5% in those without COPD.[22] The prevalence of coded osteoporosis in the GP health records presented here was, however, far lower at 5.7% than the reported prevalence from clinical studies when osteoporosis and BMD are systematically assessed. This raises the question of subclinical, undiagnosed osteoporosis leading to a missed opportunity for intervention and strengthening the need for a systematic assessment, especially when cost-efficient anti-resorptive treatment is available.[24]

There is growing consensus on COPD being a secondary cause of osteoporosis, including within the NICE clinical guidelines on osteoporosis where fracture risk prediction tools are recommended, yet in practice seem rarely done.[6] While osteoporosis in itself leads to pain and poor quality of life,[25] ultimately osteoporosis treatment aims to reduce the risk of fracture.[24 26] Risk factors for fracture include osteoporosis but also falls, which, are greater in patients with COPD.[11 27] While the increased risk of fractures in COPD has previously been considered,[28] they have not assessed the incidence from the time of COPD diagnosis or only reported as part of a larger study of post-menopausal women[29] or analysed the history of obstructive airway disease (COPD and asthma together) before the index date of osteoporotic fracture in both cases and controls over the age of 18 years.[30]

**Table 3** Discrimination measures for FRAX and QFracture at recommended treatment cut-offs for both MOF and hip fractures alone

| Discriminatory measures | FRAX | QFracture |
|---|---|---|
| | Measure for ≥20% risk (95% CI) | Measure for ≥20% risk (95% CI) |
| **All MOF** | | |
| Sensitivity | 25.4% (22.7% to 28.1%) | 25.2% (22.5% to 27.9%) |
| Specificity | 92.6% (91.0% to 94.2%) | 87.7% (85.7% to 89.7%) |
| Positive predictive value | 18.8% (16.4% to 21.1%) | 12.2% (10.2% to 14.2%) |
| Negative predictive value | 94.8% (93.4% to 96.2%) | 94.5% (93.1% to 95.9%) |
| | **Measure for ≥3% risk** | **Measure for ≥3% risk** |
| Hip fracture | | |
| Sensitivity | 78.1% (75.6% to 80.7%) | 82.1% (79.7% to 84.5%) |
| Specificity | 60.8% (57.8% to 63.8%) | 55.2% (52.1% to 58.3%) |
| Positive predictive value | 3.9% (2.7% to 5.1%) | 3.6% (2.5% to 4.8%) |
| Negative predictive value | 99.3% (98.8% to 99.8%) | 99.3% (98.8% to 99.8%) |

MOF, major osteoporotic fractures.

A

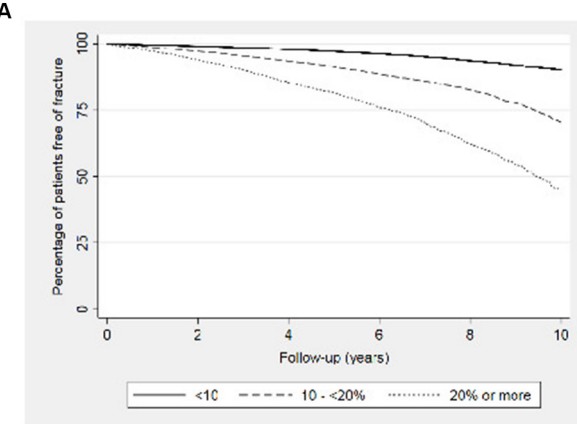

B

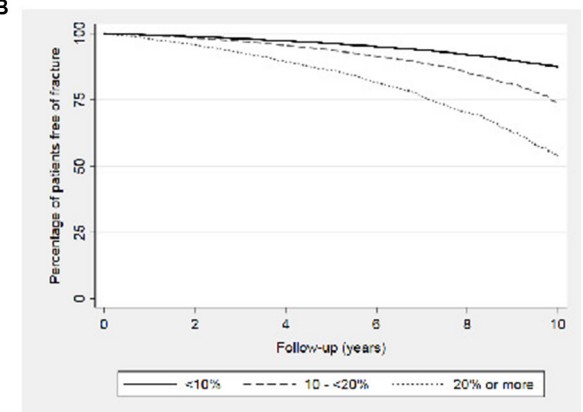

**Figure 1** Kaplan-Meier plots comparing the incidence of MOF at various predicted fracture risk categories in patients with COPD using (A) FRAX and (B) QFracture. COPD, chronic obstructive pulmonary disease; MOF, major osteoporotic fractures.

Little is known about the use of fracture risk assessment tools in patients with COPD. A number of validation studies have performed independent assessments to predict subsequent fracture in the general population.[31 32] The studies differ widely in sample size, methodology, and techniques used to assess performance.[33] Discrimination for FRAX (without BMD incorporation) and QFracture have both been reported as good.[31 34 35] The results from this COPD study are comparable to the general population validation studies;[31 34 35] however, the area under curve for MOF using QFracture was lower than that reported in other studies. Similar to findings from studies based on general population, the discrimination from our study was better in women than men and better for hip fracture than MOF.[36] The discrimination appeared similar within the 50–90-year-old group when compared with the 40–90-year olds. Despite the two tools having differences in their approach to calculating fracture risks, both predict fractures satisfactorily in patients with COPD. Despite the sensitivity and positive predictive values being far from ideal, sensitivity reported in our study are comparable to those published in studies using a general population.[31 35] Although a bespoke

COPD tool could be adapted in the future, the use of one of the established fracture risk scores in the meantime provides the opportunity to systematically identify and intervene. Such tools are incorporated into primary care medical record systems and utilised in a number of other disease areas. The available fracture prevention therapy (anti-resorptive agents) are very effective, safely yielding 40%–60% reduction in the risk of fracture.[26] These medications are cost-effective in high-risk patients—reduces morbidity, mortality and the healthcare cost associated with osteoporotic fractures.[24] The fracture prediction tools could be integrated into COPD annual assessments or at COPD diagnosis. The identification of patients at high risk is a valuable information to guide and optimise treatment options, though the optimal pathways for this integration is required.

The use of OCS has been considered to be a major contributory factor in the development of osteoporosis. However, osteoporosis has been reported in patients with no OCS use.[4 5] Other known osteoporosis risk factors are also likely to contribute in patients with COPD, including smoking, a low BMI, physical inactivity and systemic inflammation. Some of these risk factors could be moderated through education, smoking cessation, pulmonary rehabilitation and lifestyle changes.[37 38] Recognition of the scale and impact of fracture risk draws further necessary attention to these interventions to aim to prevent and reduce risks, alongside appropriate pharmacotherapy.

The study had several strengths in its methods, analyses, findings and implications for clinical practice. First, this research was population-based and compared patients with COPD with age-sex-matched control patients from the same GP. Its external validity and hence generalisability was high because the THIN database is representative of the UK population.[14] There was a substantial duration of follow-up. A wide range of potential confounders were also evaluated and adjusted for in the analyses.

For the assessment of the fracture prediction tools, the population of patients with COPD used was large, with many fracture events, and included both men and women. This minimised the likelihood of a selection bias. The assessments of the prediction tools were done using the same population, therefore minimising the effect of confounding for a difference in performance. We are presently not aware of studies that have determined the performance of the recommended fracture prediction tools in the sub-population of patients with COPD. The dataset was using UK electronic health records but is likely representative of other countries in representing the scale of the problem and the utility of the risk prediction scores.

Regarding limitations, some variables might be subject to information or reporting bias as READ codes recorded in databases do not capture free text from consultations. Such variables include patient-reported alcohol intake, use of cigarettes or their awareness of relevant family history. The possibility of residual confounding can also not be excluded as risk factors such as physical activity,

diet and ethnicity could not be adjusted for in the analyses. An accepted definition of fracture types was used; however, it is difficult to determine the cause of fracture based simply on the fracture site, with no additional information. Unlike studies which assess BMD systematically, this is not currently done in clinical practice, nor are the fracture risk scores routinely calculated as highlighted here. Therefore, the incidence of osteoporosis based on clinical codes likely reflects an underestimation of the true increased incidence/risk of osteoporosis.

In summary, despite validated fracture risk prediction tools, there was very little assessment of the increased fracture risk in patients with COPD. However, on retrospective calculation of fracture risk, the tools identify those patients with COPD at greatest risk of fracture. The identification with a systematic assessment of bone health and addressing prevention and treatment of those at a greater risk of fracture have the potential to improve outcomes for patients with COPD.

**Twitter** @Bolton_char @COPDNotts @NottmBRCLung

**Acknowledgements** We are thankful to J Hippisley-Cox and S Hippisley-Cox for the use of QFracture.

**Contributors** CEB, TMM and JES designed the study concept and design and are grant holders. RKA conducted the main statistical analysis and wrote the first draft of the manuscript. JG prepared the THIN data extracts used and assisted with the analysis. All authors contributed to the interpretation of the data, writing of the manuscript and critical revisions. CEB is a guarantor.

**Funding** This study was funded by a COPD ' Open Air ' research grant from Pfizer. CEB and TMM are supported by the NIHR Nottingham BRC. The views expressed are those of the authors and not necessarily those of the NHS, the NIHR or the Department of Health.

**Competing interests** All authors have completed the Unified Competing Interest form (available on request from the corresponding author) and declare the following: CEB, TMM and JES received an investigator-sponsored study grant from Pfizer for the submitted work; CEB reports grants from MRC/Association of British Pharmaceutical Industry, TSB, GSK and other support from Chiesi and Boehringer, outside the submitted work; JES reports personal fees from Astra Zeneca, Boehringer-Ingelheim, Nutricia, Chiesi, Sandoz, Novartis, Pfizer, MIMS, RCGP, Cogora and other support from PCRS-UK, Education for Health, Teva and NICE, outside the submitted work; and no financial relationship with any organisation that might have an interest in the submitted work in the previous three years, no other relationship or activity that could appear to have influenced the submitted work.

**Ethics approval** The study was approved by an independent Scientific Review Committee (SRC), 16THIN029.

**Provenance and peer review** Not commissioned; externally peer reviewed.

**Data sharing statement** No additional unpublished data from the study are available.

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
