## [Reviewer comments · BMJ Open]

BMJ Open

BMJ Open is committed to open peer review. As part of this commitment we make the peer review history of every article we publish publicly available.

When an article is published we post the peer reviewers' comments and the authors' responses online. We also post the versions of the paper that were used during peer review. These are the versions that the peer review comments apply to.

The versions of the paper that follow are the versions that were submitted during the peer review process. They are not the versions of record or the final published versions. They should not be cited or distributed as the published version of this manuscript.

BMJ Open is an open access journal and the full, final, typeset and author-corrected version of record of the manuscript is available on our site with no access controls, subscription charges or pay-per-view fees (<http://bmjopen.bmj.com>).

If you have any questions on BMJ Open's open peer review process please email editorial.bmjopen@bmj.com

BMJ Open

Predicting Fracture Risk in Patients with COPD using The Health Improvement Network (THIN)

Journal:	BMJ Open
Manuscript ID	bmjopen-2018-024951
Article Type:	Research
Date Submitted by the Author:	27-Jun-2018
Complete List of Authors:	Akyea, Ralph; University of Nottingham, Nottingham Respiratory Research Unit, NIHR Nottingham Biomedical Research Centre; University of Nottingham, Division of Epidemiology & Public Health McKeever, Tricia; University of Nottingham, Nottingham Respiratory Research Unit, NIHR Nottingham Biomedical Research Centre; University of Nottingham, Division of Epidemiology & Public Health Gibson, Jack; University of Nottingham, Division of Epidemiology & Public Health Scullion, Jane; Institute for Lung Health, University Hospitals of Leicester Glenfield Site Bolton, Charlotte; University of Nottingham, Nottingham Respiratory Research Unit, NIHR Nottingham Biomedical Research Centre
Keywords:	Fracture, COPD, fracture risk prediction tool, osteoporosis

Predicting Fracture Risk in Patients with COPD using The Health Improvement Network (THIN)

Ralph K Akyea^{1, 2}; Tricia M McKeever^{1, 2}; Jack E Gibson²; Jane E Scullion³; Charlotte E Bolton¹

Authors affiliations:

¹ Nottingham Respiratory Research Unit, NIHR Nottingham Biomedical Research Centre, School of Medicine, University of Nottingham, UK.

² Division of Epidemiology and Public Health, School of Medicine, University of Nottingham, UK.

³ University Hospitals of Leicester Glenfield Site, Institute for Lung Health, Leicester, UK.

Corresponding author information:

Dr Charlotte E Bolton,

Nottingham Respiratory Research Unit, NIHR Nottingham Biomedical Research Centre, School of Medicine, University of Nottingham, City Hospital Campus, Hucknall Road, Nottingham, NG5 1PB, UK.

Tel: +44 (0)115 8231710

Email: charlotte.bolton@nottingham.ac.uk

Keywords:

Fracture, osteoporosis, COPD, fracture risk prediction tool

Word counts:

Abstract: 259

Main text: 2,795

ABSTRACT

Objectives To assess incidence of hip fracture or major osteoporotic fractures (MOF) in patients with COPD compared to non-COPD subjects and to evaluate the use and performance of fracture risk prediction tools in patients. To assess recorded osteoporosis diagnosis.

Design A population-based cohort study

Setting UK General Practice health records from The Health Improvement Network database

Participants Patients with an incident COPD diagnosis from 2004-2015 and age, sex and general practice matched non-COPD subjects were studied.

Outcomes Incidence of fracture; accuracy of fracture risk prediction tools in COPD; prevalence and incidence of osteoporosis.

Methods: Stratified Cox proportional hazards models (stratified matched cohort analyses) were used. The discriminatory accuracy (area under the receiver operating curve [ROC]) of fracture risk prediction tools in COPD was assessed.

Results There was an increased risk of fracture in patients with COPD but this was largely mediated through oral corticosteroid use, BMI and smoking. Retrospectively calculated discriminatory accuracies for major osteoporotic fracture were FRAX[®]: 71.4% (95% CI: 70.6 to 72.2%), QFracture[®]: 61.4% (95% CI: 60.5 to 62.3%) and for hip fracture both 76.1% (95% CI: 74.9 to 77.2%). Prevalence of osteoporosis was greater for patients (5.7%) compared to non-COPD subjects (3.9%), $p < 0.001$. In those without former osteoporosis, patients ($n = 73,084$) had an increased osteoporosis incidence compared to non-COPD subjects ($n = 264,544$), (adjusted hazard ratio, 1.13, 95% CI 1.05 to 1.22).

Conclusion COPD patients are at increased risk of fractures and osteoporosis. Despite this, there is no systematic assessment of fracture risk in clinical practice. Fracture risk tools identify those at high-risk of fracture in patients with COPD.

Strengths and limitations of this study

- This research was population-based using electronic health records representative of the UK population with a substantial duration of follow-up.
- A wide range of potential confounders were also evaluated and adjusted for in the analyses.
- For the assessment of the fracture prediction tools, the population of patients with COPD used was large, with many fracture events, included both men and women and is representative of the UK population.
- Whilst coded osteoporosis diagnosis appears under-reported in COPD compared to where osteoporosis is systematically sought in patients with COPD, this was a secondary outcome. Further the under-reporting is worthy of mention.

INTRODUCTION

Osteoporosis in both male and female patients with COPD is firmly established as one of the core comorbid conditions.[1,2] Over the last decade, it has become clear that osteoporosis is not just an end-stage COPD problem[3] nor just in those on maintenance oral corticosteroids, but it also occurs in a large proportion of those with mild-moderate airflow obstruction and even in steroid naïve patients.[4,5] The Global Initiative for Chronic Obstructive Pulmonary Disease (GOLD strategy recommends that osteoporosis co-existence should be considered in COPD [1] and the UK National Institute for Health and Care Excellence (NICE) Guidelines on osteoporosis considers COPD as a secondary cause of osteoporosis encouraging use of fracture prediction tools.[6]

The causes for osteoporosis in COPD are likely multiple and cumulative, including age, smoking exposure, inactivity, low body mass index (BMI), systemic inflammation and the frequent use of oral corticosteroids.[7] The clinical implications of osteoporosis include increased risk of fractures, poor quality of life, pain and further deterioration in lung function.[8,9] Osteoporosis can also remain undiagnosed as asymptomatic for many years.[10] Fractures are a function of trauma sustained, such as falls which are common in COPD [11], and the quality and architecture of bone. Fractures contribute further pain, poor quality of life, increased mortality and confer a substantial economic burden on health systems, patients and their families.[12,13] Given this, the individual risk of a future fracture in patients with COPD is crucial to determine in patient care and to treat accordingly.

Fracture risk prediction tools based on clinical and personal characteristics have been developed over the years to guide investigation and management of those identified to be at high risk of osteoporotic fractures, worldwide. These include for the UK (and many other regions), FRAX[®] and QFracture[®]. [6]

The full extent of fracture risk assessment in patients with COPD is not fully established. The aim of this study was to evaluate fracture in patients with COPD compared to non-COPD subjects together with the use of and the performance of fracture risk prediction tools in patients with COPD. Further, to assess the coding of osteoporosis in patients with COPD and non COPD subjects.

METHODS

Information for this cohort study was obtained from The Health Improvement Network (THIN), an anonymised primary care database representing 6.2% of the total UK population.[14]

The study population consisted of patients 40 years and over with a new Read coded COPD diagnosis during the data collection period 1/1/04-31/12/15, with at least 1 year of record prior to COPD diagnosis.[15] Each patient was matched by age, gender and GP practice to up to four subjects without a history of COPD to generate a matched cohort and assigned the same index date.

Follow up was from the index date to the first record of either the occurrence of the outcome of interest (fracture/osteoporosis), the date of transfer of the patient out of the practice area, death or end of THIN data collection. Read coded diagnoses for osteoporosis and read coded hip fracture or major osteoporotic fractures (MOF) (fracture of the hip, proximal humerus, forearm or clinically symptomatic vertebra/spine) were ascertained.

A series of explanatory variables [6,16] determined at baseline (prior to or at index date) included: Charlson Comorbidity Index (CCI) score,[17] Townsend social deprivation score, fall, prior fractures, parental history of fall/osteoporosis, relevant comorbidities and secondary causes of osteoporosis as defined in the FRAX[®] questionnaire.[18] Records for smoking status, alcohol use, MRC Dyspnoea scale, BMI, and use of specific prescription drugs were restricted to a defined time period. Oral corticosteroid (OCS) use was considered as a time-dependent variable with exposed and non-exposed periods. Exposed periods started from prescription date until the first gap of more than 90 days between prescriptions. OCS prescriptions issued within 90 days prior to index date were considered as part of exposed periods.

Input variables included clinical status, prescription drug use, and demographic characteristics, according to the variables/definitions used in both FRAX[®] and QFracture[®] tools,[18,19], additional detail on the method is provided in an online data supplement (*Appendix 1*). Imputation was used for missing variables.

The 10-year risk score for hip fracture and MOF according to QFracture[®] (version
2017.0.0.0) and FRAX[®] for UK without BMD information (desktop version 3.12) were
calculated for patients with COPD, aged 40-90 years old.

9 **Statistical analyses**

Incidence rates were calculated for both groups using time-dependent Cox
proportional hazards regression to estimate hazard ratios (HRs) of osteoporosis and
fracture risks, with OCS treated as a time-dependent variable. Confounders were
included in the final model when independently changing the HRs for
osteoporosis/fracture by at least 5%. A former osteoporosis diagnosis or
antiresorptive treatment prior to COPD diagnosis excluded that subject from analyses
related to either osteoporosis incidence or risk. In addition to evaluating incidence in
the whole cohort, separate sub-analyses excluded a) patients with COPD and no
documented smoking history together with their matched non-COPD subjects and b)
those with no prior record of osteoporosis.

To evaluate FRAX[®] and QFracture[®], the outcome was treated as a binary variable
(fracture or no fracture). Fracture risk probabilities were categorised based on
recommended treatment thresholds ($\geq 20\%$ for MOF and $\geq 3\%$ for hip fracture).[20]
To evaluate the overall ability of each tool to discriminate (performance) between
those at low and high risks, the area under the receiver operating characteristic
(ROC) curve was calculated. Sensitivity, specificity, positive and negative predictive
values were calculated. Survival analysis was performed and Kaplan-Meier plots
comparing the fracture incidence were generated.

All statistical analyses were performed using Stata 15.0 (StataCorp LP).

**Patient involvement**

The results and implications of previous research from the team on systematic
assessment of osteoporosis in patients with COPD [4] has been discussed extensively
in previous patient meetings. Whilst this and other literature has strengthened the
GOLD strategy recommendations,[1] evaluation of clinical services would suggest
systematic assessment is not done in patients. More recently, patients with COPD
out-patient clinics have approached the principal investigator at the time of their "ad
hoc osteoporosis" diagnosis to ask why this was not investigated at or closer to
COPD diagnosis and how osteoporosis could be assessed. This has led to the
development of this grant application with significant patient input in the design and

context. The results have been discussed back with representatives on the
respiratory research panel. Given the implications for clinical practice, the findings
have been discussed extensively at the PPI meeting and a Breathe Easy meeting in
early 2018. A lay summary has been developed for the patient newsletter (n>700)
and website. In the meantime, members of the respiratory research panel are
assisting the PI in planning future work regarding implementation.

For peer review only

RESULTS

The baseline characteristics are shown in Table 1. A total of 80,874 eligible patients with COPD and 308,999 matched non-COPD subjects were identified. The median follow-up period was 5 years for both patients and non-COPD subjects.

Osteoporosis at index date and incidence

Within 1 year (before and after) of the index date, 1,504 (1.86%) patients with COPD had a new recorded diagnosis of osteoporosis compared to 3,059 (1.12%) in matched non-COPD subjects, $p < 0.001$. 3,186 (3.94%) of patients with COPD had a diagnosis of osteoporosis more than a year prior to index date compared to 8,822 (2.86%) for the matched controls $p < 0.001$.

1,457 (1.80%) patients with COPD compared to 3,694 (1.20%) non-COPD subjects, had a record of any diagnostic assessment for osteoporosis, recorded within 1 year (before and after) of the index date, ($p < 0.001$).

Demographics remained similar after excluding those with former osteoporosis. Patients with COPD ($n=73,084$) compared to non-COPD subjects ($n=264,544$) were significantly more likely to have incident diagnosis of osteoporosis (crude hazard ratio (HR), 1.96; 95% confidence interval [CI] 1.87 to 2.05; *Appendix 2*).

Incidence of Fracture

There was a significantly increased risk of both MOF, crude hazard ratio of 1.60 (95% CI 1.52 to 1.69) and hip fractures: 1.67 (95% CI 1.56 to 1.80) in patients with COPD compared to non-COPD subjects in the unadjusted model, which remained significant after adjustment for age, gender and GP practice. In the fully adjusted models the association were diminished (Table 2). Smoking status altered the effect between COPD and fracture the most, followed by BMI, CCI score and oral corticosteroid.

Sensitivity analysis with participants with no former osteoporosis showed similar results. The risk of major osteoporosis fracture was also similar when evaluated in only patients with COPD with a prior history of smoking and their matched controls. However, here, the risk of hip fracture remained significantly increased in the

adjusted model compared to non-COPD subjects (aHR, 1.13; 95% CI 1.004 to
1.280; p-value: 0.043).

**Fracture risk prediction tools in COPD**

Only 1074 (1.33%) of patients with COPD had a FRAX[®] assessment READ coded
ever documented in the records and 12 patients had a READ coded QFracture[®]
assessment. Within 1 year (before and after) of index date, 248 (0.31%) of patients
with COPD had a FRAX[®] and only 1 patient a QFracture[®].

The final population for the discriminatory accuracy analysis comprised 72,559
patients aged 40-90 years with COPD and no prior diagnosis of osteoporosis or
prescription of any anti-resorptive treatment (*demographics in Appendix 3*). This
included 4,605 (6.4%) who experienced a MOF and 1,444 (2.0%) who experienced
hip fracture.

When the FRAX[®] and QFracture[®] scores were calculated for patients with COPD, for
hip fracture 29,035 (40.0%) had a risk $\geq 3\%$ using FRAX[®] and 33,065 (45.6%) using
the QFracture[®]. For MOF, 6,221 (8.6%) of patients had a risk $\geq 20\%$ using FRAX[®] and
9,546 (13.2%) using QFracture[®].

Both risk tools had a similar discriminatory accuracy for hip fracture (FRAX[®] 76.1%,
95% CI 74.9 to 77.2% and QFracture[®] 76.1%, 95% CI 74.9 to 77.2%). FRAX[®],
however, had a higher accuracy for MOF (71.4% 95% CI 70.6% to 72.2%) than
QFracture[®] (61.4% 95% CI 60.5% to 62.3%).

The discriminatory accuracies were better in women than men. The performance of
the prediction tools was similar in the patients aged 50-90 years compared to 40-90-
40 year olds. Table 3 shows the results for the sensitivity, specificity, positive and
41 negative predictive values assessed for the performance of the prediction tools at
42 $\geq 3\%$ risk probability for hip fracture and $\geq 20\%$ risk probability for major
osteoporotic fractures.

The Kaplan-Meier plots for time to first MOF for QFracture[®] and FRAX[®] are presented
in Figure 1.

DISCUSSION

Using UK primary care electronic health records, we have reported on the burden of fractures in patients with COPD with both hip and major osteoporotic fractures increased in patients with COPD compared to age, gender and GP surgery matched subjects. Despite the increased fracture risk and recommendations in the NICE osteoporosis guidelines, fracture risk prediction tools are rarely coded. However, where the risk score was retrospectively calculated, the risk prediction tools identify those at risk of hip fracture or MOF. Therefore, fracture risk prediction and subsequent targeted therapy and management could transform multi-morbidity management of COPD. In addition, we report that the prevalence and incidence of osteoporosis, a risk for fracture, in patients with COPD, is far greater than in non-COPD subjects.

Prevalence of osteoporosis varies widely in the different studies of patients with COPD. This is mainly dependent on the severity of COPD,[4,5] whether osteoporosis was systematically sought or self-reported [4,21], and whether patients included were on oral corticosteroids.[3] A prevalence of 23-32% has been reported where BMD was systematically performed [22].[4], while 14% of patients with COPD self-reported osteoporosis compared to 5% in those without COPD.[21] The prevalence of coded osteoporosis in the GP health records was, however, far lower at 5.7% than the reported prevalence from clinical studies when osteoporosis and BMD are systematically assessed. This raises the question of subclinical, undiagnosed disease leading to a missed opportunity for intervention and strengthening the need for a systematic assessment especially when cost-efficient anti-resorptive treatment is available.[23]

There is growing consensus on COPD being a secondary cause of osteoporosis, including within the NICE clinical guideline on osteoporosis where fracture risk prediction tools are recommended, yet in practice seem rarely done.[6] Whilst osteoporosis in itself leads to pain and poor quality of life,[24] ultimately osteoporosis treatment aims to reduce the risk of fracture.[23,25] Risk factors for fracture include osteoporosis but also falls, which, are greater in patients with COPD.[11,26] Whilst the increased risk of fractures in COPD has previously been considered,[27] they have not assessed incidence from time of COPD diagnosis or only reported as part of a larger study of post-menopausal women [28] or analysed the history of obstructive airway disease (both COPD and asthma together) before

the index date of osteoporotic fracture in both cases and controls over the age of 18
4 years.[29]

Little is known about the use of fracture risk assessment tools in patients with COPD.
A number of validation studies have performed independent assessments to predict
subsequent fracture in the general population.[32,33] The studies differ widely in
sample size, methodology, and techniques used to assess performance.[34]
Discrimination for FRAX[®] (without BMD incorporation) and QFracture[®] have both
been reported as good.[32,35,36] The results from this COPD study are comparable
to the general population validation studies.[32,35,36] The discrimination from our
study was better in women and for hip fracture as it is in the general population
studies – both associated with the greatest morbidity and mortality.[37] The
discrimination appeared similar within the 40-90 and 50-90 year-old groups. Despite
the two tools having differences in their approach to calculating fracture risks, both
predict fractures satisfactorily in patients with COPD and will thus be helpful in
selecting high-risk patients. Available fracture prevention therapy (anti-resorptive
agents) are very effective, safely yielding 40-60% reduction in the risk of
fracture.[25] These medications are cost-effective in high-risk patients –reduces
morbidity, mortality and health care cost associated with osteoporotic fractures.[23]
These fracture prediction tools could be integrated into COPD annual assessments or
diagnosis to identify patients at high fracture risk, assist in selecting efficacious
treatment and provide long-term follow-up with serial assessments. Though the
optimal pathways for this integration is required.

The use of oral corticosteroids has been considered to be a major contributory factor
in the development of osteoporosis. However, osteoporosis has been reported in
patients with no oral corticosteroid use.[4,5] Other known osteoporosis risk factors
are also likely to contribute in patients with COPD including smoking, a low BMI,
physical inactivity and systemic inflammation. Some of these risk factors could be
moderated through education, smoking cessation, pulmonary rehabilitation and
lifestyle changes.[30,31] Recognition of the scale and impact of fracture risk draws
further necessary attention to these interventions to aim to prevent and reduce risks,
alongside appropriate pharmacotherapy.

The study had several strengths in its methods, analyses, findings, and implications
for clinical practice. Firstly, this research was population-based and compared
patients with COPD with age-sex matched control subjects from the same general
practice. Its external validity and hence generalisability was high because THIN

database is representative of the UK population. There was a substantial duration of
follow-up. A wide range of potential confounders were also evaluated and adjusted
for in the analyses.

For the assessment of the fracture prediction tools, the population of patients with
COPD used was large, with many fracture events, included both men and women and
is representative of the UK population. This minimised the likelihood of a selection
bias. The assessments of the prediction tools were done using the same population,
therefore minimising the effect of confounding for a difference in performance. We
are presently not aware of studies that have determined the performance of the
recommended fracture prediction tools in the sub-population of patients with COPD.
The dataset was using UK electronic health records but is likely representative of
other countries in representing the scale of the problem and the utility of the risk
prediction scores.

Regarding limitations, some variables might be subject to information or reporting
bias, including patient reported alcohol intake, use of cigarettes or their awareness of
relevant family history. The possibility of residual confounding can also not be
excluded as risk factors such as physical activity, diet and ethnicity could not be
adjusted for in the analyses. An accepted definition of fractures types was used;
however, it is difficult to determine the cause of fracture based simply on fracture
site, with no additional information. Unlike studies which assess BMD systematically,
this is not currently done in clinical practice, nor are the fracture risk scores routinely
calculated as highlighted here. Therefore, the incidence of osteoporosis based on
clinical codes likely reflects an underestimation of the true increased incidence/risk of
osteoporosis.

In summary, despite validated fracture risk prediction tools, there was very little
assessment of the increased fracture risk in patients with COPD. However, on
retrospective calculation of fracture risk, the tools identify those patients with COPD
at greatest risk of fracture. Identification with a systematic assessment of bone
health and addressing prevention and treatment of those at greatest risk of fracture
would improve quality of life and outcomes for patients with COPD.

Acknowledgements

With grateful thanks to Prof J Hippisley-Cox and Mr S Hippisley-Cox for use of the QFracture®.

Competing interests

All authors have completed the Unified Competing Interest form (available on request from the corresponding author) and declare: CEB, TMM, JES received an investigator sponsored study grant from Pfizer for the submitted work; CEB reports grants from MRC/Association of British Pharmaceutical Industry (ABPI), TSB, GSK and other support from Chiesi and Boehringer, outside the submitted work; JES reports personal fees from Astra Zeneca, Boehringer-Ingelheim, Nutricia, Chiesi, Sandoz, Novartis, Pfizer, MIMS, RCGP, Cogora and other support from PCRS-UK, Education for Health, Teva and NICE outside the submitted work; no financial relationships with any organisations that might have an interest in the submitted work in the previous three years, no other relationships or activities that could appear to have influenced the submitted work.

Author contributions

CEB, TMM and JES designed study concept and design and are grant holders. RKA conducted the main statistical analysis and wrote the first draft of the manuscript. JEG prepared the THIN data extracts used and assisted with analysis. All authors contributed to the interpretation of the data, writing of the manuscript and critical revisions.

CEB is guarantor.

Ethical approval

The study was approved by an independent Scientific Review Committee (SRC), 16THIN029.

Funding

This study was funded by a COPD "Open Air" research grant from Pfizer.

The funders had no role in study design, data collection and analysis, decision to publish or preparation of the manuscript. CEB and TMM are supported by the NIHR Nottingham BRC. The views expressed are those of the authors and not necessarily those of the NHS, the NIHR or the Department of Health.

Data sharing

No additional data available.

REFERENCES

[revised manuscript text omitted]

Table 1: Baseline characteristics of patients with COPD and non-COPD subjects

Descriptor	COPD patients		Non-COPD subjects		p-value
	n = 80,874	%	n = 308,999	%	
Mean age at index date (years, SD)	66.9 (11.0)		66.5 (10.9)		
Gender					0.002
Male	42,799	52.9	161,648	52.3	
Female	38,075	47.1	147,351	47.7	
Follow-up (years, median, IQR)	5.28	2.6-8.3	5.24	2.6-8.3	
MRC Dyspnoea Scale (1 Year either side of diagnosis)					<0.001
1	9,499	11.8	1,168	0.4	
2	19,466	24.1	1,092	0.4	
3	10,488	13.0	446	0.1	
4 & 5	5,237	6.5	177	0.1	
No record	36,184	44.7	306,116	99.1	
Charlson Comorbidity Index Score					<0.001
0	0	0.0	172,566	55.9	
1	41,777	51.7	50,955	16.5	
2	13,506	16.7	42,667	13.8	
3	12,694	15.7	23,546	7.6	
≥ 4	12,897	16.0	19,265	6.2	
Body Mass Index (BMI) (kg/m²)					<0.001
Underweight (< 18.5)	3,414	4.2	2,699	0.9	
Normal (18.5 – 24.9)	24,734	30.6	54,267	17.6	
Overweight (25 – 29.9)	23,497	29.1	77,129	25.0	
Obese (≥30)	19,083	23.6	60,280	19.5	
No BMI	10,146	12.6	114,624	37.1	
Smoking status (1 Year either side of diagnosis)					<0.001
Never smoked	7,925	9.8	94,800	30.7	
Ex-smoker	38,590	47.7	72,989	23.6	
Current smoker	32,436	40.1	34,691	11.2	
Unknown	1,923	2.4	106,519	34.5	
History of Falls (prior to or at diagnosis)					
Personal history	8,969	11.1	26,203	8.5	<0.001
Parental history of fall/osteoporosis	96	0.1	298	0.1	0.076
Medications (1 Year either side of diagnosis)					
Oral Glucocorticoid Use	33,618	41.6	19,479	6.3	<0.001
Inhaled Corticosteroid Use	47,574	58.8	21,312	6.9	<0.001

Table 2: Risk of fractures in patients with COPD compared with non-COPD subjects

	Number of fractures	Rate/1,000 person-years	Crude HR (95% CI)	Fully adjusted HR (95% CI)
Major osteoporotic fractures				
Non-COPD subjects	6,032	4.32	Reference	Reference
Patients with COPD	2,234	6.64	1.60 (1.52 – 1.69)	1.04 (0.96 – 1.12) ^a
Hip fracture				
Non-COPD subjects	3,170	2.26	Reference	Reference
Patients with COPD	1,213	3.57	1.67 (1.56 – 1.80)	1.09 (0.98 – 1.21) ^b

HR – Hazard ratio; CI – Confidence interval

Crude HR – Cox regression model derived HR adjusted for age, sex, and GP practice

^a Multivariate Cox regression model derived HR was adjusted for age, sex, GP practice, Charlson Comorbidity Index, Body Mass Index, smoking status, inhaled corticosteroid use, antidepressant use and cumulative oral corticosteroid use.

^b Multivariate Cox regression model derived HR was adjusted for age, sex, GP practice, Charlson Comorbidity Index, Body Mass Index, smoking status, inhaled corticosteroid use and cumulative oral corticosteroid use.

Table 3: Discrimination measures for FRAX® and QFracture® at recommended treatment cut offs for both major osteoporotic and hip fractures

Discriminatory measures	FRAX®	QFracture®
	Measure for $\geq 20\%$ risk	Measure for $\geq 20\%$ risk
Major Osteoporotic fractures		
Sensitivity	25.4%	25.2%
Specificity	92.6%	87.7%
Positive Predictive Value	18.8%	12.2%
Negative Predictive Value	94.8%	94.5%
	Measure for $\geq 3\%$ risk	Measure for $\geq 3\%$ risk
Hip fracture		
Sensitivity	78.1%	82.1%
Specificity	60.8%	55.2%
Positive Predictive Value	3.9%	3.6%
Negative Predictive Value	99.3%	99.3%

**Figure 1: Kaplan-Meier plots comparing incidence of major osteoporotic**
**fractures at various predicted fracture risk categories in patients with COPD**
**using (a) FRAX® and (b) QFracture®**

For peer review only

Figure 1: Kaplan-Meier plots comparing incidence of major osteoporotic fractures at various predicted fracture risk categories in patients with COPD using (a) FRAX® and (b) QFracture®

90x153mm (300 x 300 DPI)

Appendix 1

METHODS

Potential confounders

For smoking status, alcohol use, MRC Dyspnoea scale, and a list of prescription drugs, the most recent record within 1 year (before and after) of index date were used. A BMI record within 2 years (before and after) of index date was used.

Where possible BMI was calculated from height and weight records, for patients with a missing BMI record. The BMI was subsequently categorised (underweight: <18.5 kg/m², normal: 18.5 - <25 kg/m², overweight: 25 - <30 kg/m², obese: >30 kg/m²).

Having received at least one prescription for inhaled corticosteroids, anti-epileptics, antidepressants, oestrogen-only Hormone Replacement Therapy (HRT) and osteoporosis medications, within 1 year (before and after) of index date were considered as risk factors.

Prediction tools – Input variables

The respective variable definitions as outlined in the algorithms for the prediction tools were used.

Smoking status – In QFracture[®], three current smoking categories are provided according to the number of cigarettes smoked daily[1]. To avoid the bias of categorising patients in one of the outlying categories, “current smokers” with no documented number of cigarettes smoked were assigned to the middle category “10-19 cigarettes daily” as done in a recent publication [2]. For FRAX[®]’s two-category smoking status, former smokers were assigned to the “non-smoker” category as was done in the cohorts used to develop FRAX[®]. [3]

Alcohol consumption – similarly, for alcohol use in QFracture[®], alcohol drinkers with no documented unit/day intake were assigned to “moderate (3-6units/day)”.

Missing values for BMI, smoking status, and alcohol use were imputed by multiple imputation using all predictors, resulting in twenty imputed datasets. A complete case sensitivity analysis without imputed variables was also performed.

References

- 1 ClinRisk Ltd. QFracture-2016® risk calculator. <http://www.qfracture.org/> (accessed 20 Sep 2017).
- 2 Dagan N, Cohen-Stavi C, Leventer-Roberts M, *et al*. External validation and comparison of three prediction tools for risk of osteoporotic fractures using data from population based electronic health records: retrospective cohort study. *BMJ* 2017;**356**:i6755. doi:10.1136/BMJ.I6755
- 3 Kanis JA, Oden A, Johnell O, *et al*. The use of clinical risk factors enhances the performance of BMD in the prediction of hip and osteoporotic fractures in men and women. *Osteoporos Int* 2007;**18**:1033–46. doi:10.1007/s00198-007-0343-y

Appendix 2

Table E1: Risk of osteoporosis in patients with COPD compared with non-COPD subjects

Descriptor	Crude HR (95% CI)	Fully adjusted HR (95% CI)
COPD		
Non-COPD subjects	Reference	Reference
COPD patients	1.96 (1.87 – 2.06)	1.13 (1.05 – 1.22)
Charlson Comorbidity Index		
Score 0	Reference	Reference
Score 1	1.27 (1.18 – 1.36)	1.14 (1.06 – 1.23)
Score 2	1.34 (1.24 – 1.44)	1.27 (1.17 – 1.37)
Score 3	1.41 (1.28 – 1.55)	1.29 (1.17 – 1.42)
Score 4 & more	1.48 (1.33 – 1.64)	1.44 (1.29 – 1.61)
Body Mass Index (kg/m²)		
Underweight (<18.5)	1.93 (1.64 – 2.27)	1.91 (1.63 – 2.25)
Normal (18.5 – 24.9)	Reference	Reference
Overweight (25 – 29.9)	0.64 (0.60 – 0.69)	0.63 (0.58 – 0.67)
Obese (≥ 30)	0.47 (0.43 – 0.51)	0.45 (0.41 – 0.48)
No record	0.50 (0.46 – 0.53)	0.57 (0.52 – 0.61)
Smoking status		
Never	Reference	Reference
Ex	1.01 (0.95 – 1.08)	1.02 (0.95 – 1.09)
Current	1.23 (1.13 – 1.33)	1.15 (1.06 – 1.25)
Unknown	0.69 (0.64 – 0.74)	0.77 (0.71 – 0.83)
Oral Corticosteroid Use		
Unexposed	Reference	Reference
Exposed	2.79 (2.56 – 3.05)	1.91 (1.73 – 2.10)
Inhaled Corticosteroid Use		
No	Reference	Reference
Yes	1.35 (1.26 – 1.45)	1.24 (1.15 – 1.34)

HR – Hazard ratio; CI – Confidence interval

Crude HR – Cox regression model derived HR adjusted for age, sex, and GP practice

The adjusted Hazard Ratio (aHR) was 1.13, 95% CI 1.05 to 1.22, p<0.0001 – the multivariate Cox regression model derived aHR was adjusted for age, sex, GP practice, Charlson comorbidity index, body mass index, smoking status, inhaled corticosteroid use, and cumulative oral corticosteroid use.

Appendix 3

Table E2: Baseline characteristics of patients with COPD aged 40-90 years with no prior diagnosis of osteoporosis or prescription of any anti-resorptive treatment

Descriptor	COPD patients	
	n = 72,559	%
Mean age at index date (years, SD)	66.1 (10.7)	
Gender		
Female	31,885	43.9
MRC Dyspnoea Scale (1 Year either side of diagnosis)		
1	8,882	12.2
2	17,718	24.4
3	9,257	12.8
4 & 5	4,346	6.0
No record	32,356	44.6
Charlson Comorbidity Index Score		
0	0	0
1	38,573	53.2
2	11,953	16.5
3	11,110	15.3
≥ 4	10,923	15.1
Body Mass Index (BMI) (kg/m²)		
Underweight (< 18.5)	2,730	3.8
Normal (18.5 – 24.9)	21,791	30.0
Overweight (25 – 29.9)	21,504	29.6
Obese (≥30)	17,627	24.3
No BMI	8,907	12.3
Smoking status (1 Year either side of diagnosis)		
Never smoked	7,062	9.7
Ex-smoker	33,810	46.6
Current smoker	29,949	41.3
Unknown	1,738	2.4

STROBE 2007 (v4) Statement—Checklist of items that should be included in reports of *cohort studies*

Section/Topic	Item #	Recommendation	Reported on page #
Title and abstract	1	(a) Indicate the study's design with a commonly used term in the title or the abstract	1
		(b) Provide in the abstract an informative and balanced summary of what was done and what was found	2
Introduction			
Background/rationale	2	Explain the scientific background and rationale for the investigation being reported	4
Objectives	3	State specific objectives, including any pre-specified hypotheses	4
Methods			
Study design	4	Present key elements of study design early in the paper	5
Setting	5	Describe the setting, locations, and relevant dates, including periods of recruitment, exposure, follow-up, and data collection	5
Participants	6	(a) Give the eligibility criteria, and the sources and methods of selection of participants. Describe methods of follow-up	5
		(b) For matched studies, give matching criteria and number of exposed and unexposed	5
Variables	7	Clearly define all outcomes, exposures, predictors, potential confounders, and effect modifiers. Give diagnostic criteria, if applicable	5, 6
Data sources/ measurement	8*	For each variable of interest, give sources of data and details of methods of assessment (measurement). Describe comparability of assessment methods if there is more than one group	5, 6
Bias	9	Describe any efforts to address potential sources of bias	6
Study size	10	Explain how the study size was arrived at	-
Quantitative variables	11	Explain how quantitative variables were handled in the analyses. If applicable, describe which groupings were chosen and why	6
Statistical methods	12	(a) Describe all statistical methods, including those used to control for confounding	6
		(b) Describe any methods used to examine subgroups and interactions	-
		(c) Explain how missing data were addressed	6, Appendix 1
		(d) If applicable, explain how loss to follow-up was addressed	-
		(e) Describe any sensitivity analyses	6
Results			

Participants	13*	(a) Report numbers of individuals at each stage of study—eg numbers potentially eligible, examined for eligibility, confirmed eligible, included in the study, completing follow-up, and analysed	8
		(b) Give reasons for non-participation at each stage	-
		(c) Consider use of a flow diagram	-
Descriptive data	14*	(a) Give characteristics of study participants (eg demographic, clinical, social) and information on exposures and potential confounders	8, Table 1 (17)
		(b) Indicate number of participants with missing data for each variable of interest	Table 1 (17)
		(c) Summarise follow-up time (eg, average and total amount)	8
Outcome data	15*	Report numbers of outcome events or summary measures over time	8, 9
Main results	16	(a) Give unadjusted estimates and, if applicable, confounder-adjusted estimates and their precision (eg, 95% confidence interval). Make clear which confounders were adjusted for and why they were included	8, 9, Table 2 (18), Appendix 2
		(b) Report category boundaries when continuous variables were categorized	Appendix 1
		(c) If relevant, consider translating estimates of relative risk into absolute risk for a meaningful time period	-
Other analyses	17	Report other analyses done—eg analyses of subgroups and interactions, and sensitivity analyses	8, 9
Discussion			
Key results	18	Summarise key results with reference to study objectives	10
Limitations			
Interpretation	20	Give a cautious overall interpretation of results considering objectives, limitations, multiplicity of analyses, results from similar studies, and other relevant evidence	10 - 12
Generalisability	21	Discuss the generalisability (external validity) of the study results	11 & 12
Other information			
Funding	22	Give the source of funding and the role of the funders for the present study and, if applicable, for the original study on which the present article is based	13

*Give information separately for cases and controls in case-control studies and, if applicable, for exposed and unexposed groups in cohort and cross-sectional studies.

Note: An Explanation and Elaboration article discusses each checklist item and gives methodological background and published examples of transparent reporting. The STROBE checklist is best used in conjunction with this article (freely available on the Web sites of PLoS Medicine at <http://www.plosmedicine.org/>, Annals of Internal Medicine at <http://www.annals.org/>, and Epidemiology at <http://www.epidem.com/>). Information on the STROBE Initiative is available at www.strobe-statement.org.

BMJ Open

Predicting Fracture Risk in Patients with Chronic Obstructive Pulmonary Disease: A UK-based Population-based Cohort Study

Journal:	BMJ Open
Manuscript ID	bmjopen-2018-024951.R1
Article Type:	Research
Date Submitted by the Author:	17-Oct-2018
Complete List of Authors:	Akyea, Ralph; University of Nottingham, Nottingham Respiratory Research Unit, NIHR Nottingham Biomedical Research Centre; University of Nottingham, Division of Epidemiology & Public Health McKeever, Tricia; University of Nottingham, Nottingham Respiratory Research Unit, NIHR Nottingham Biomedical Research Centre; University of Nottingham, Division of Epidemiology & Public Health Gibson, Jack; University of Nottingham, Division of Epidemiology & Public Health Scullion, Jane; Institute for Lung Health, University Hospitals of Leicester Glenfield Site Bolton, Charlotte; University of Nottingham, Nottingham Respiratory Research Unit, NIHR Nottingham Biomedical Research Centre
Primary Subject Heading:	Respiratory medicine
Secondary Subject Heading:	Epidemiology
Keywords:	Fracture, COPD, fracture risk prediction tool, osteoporosis

Predicting Fracture Risk in Patients with Chronic Obstructive Pulmonary Disease: A UK-based Population-based Cohort Study

Ralph K Akyea^{1, 2}; Tricia M McKeever^{1, 2}; Jack E Gibson²; Jane E Scullion³; Charlotte E Bolton¹

Authors affiliations:

¹ Nottingham Respiratory Research Unit, NIHR Nottingham Biomedical Research Centre, School of Medicine, University of Nottingham, UK.

² Division of Epidemiology and Public Health, School of Medicine, University of Nottingham, UK.

³ University Hospitals of Leicester Glenfield Site, Institute for Lung Health, Leicester, UK.

Corresponding author information:

Dr Charlotte E Bolton,

Nottingham Respiratory Research Unit, NIHR Nottingham Biomedical Research Centre, School of Medicine, University of Nottingham, City Hospital Campus, Hucknall Road, Nottingham, NG5 1PB, UK.

Tel: +44 (0)115 8231710

Email: charlotte.bolton@nottingham.ac.uk

Keywords:

Fracture, osteoporosis, COPD, fracture risk prediction tool

Word counts:

Abstract: 283

Main text: 3,012

ABSTRACT

Objectives To assess incidence of hip fracture or major osteoporotic fractures (MOF) in patients with COPD compared to non-COPD subjects and to evaluate the use and performance of fracture risk prediction tools in patients. To assess the prevalence of osteoporosis.

Design A population-based cohort study

Setting UK General Practice health records from The Health Improvement Network database

Participants Patients with an incident COPD diagnosis from 2004-2015 and age, sex and general practice matched non-COPD subjects were studied.

Outcomes Incidence of fracture; accuracy of fracture risk prediction tools in COPD; prevalence and incidence of osteoporosis.

Methods: Cox proportional hazards models were used to assess the incidence rates of fracture and osteoporosis. The discriminatory accuracy (area under the receiver operating curve [ROC]) of fracture risk prediction tools in COPD was assessed.

Results The cohort included 80,874 eligible patients with COPD and 308,999 matched non-COPD subjects. There was an increased risk of fracture in patients with COPD but this was largely mediated through oral corticosteroid use, BMI and smoking. Retrospectively calculated discriminatory accuracies for major osteoporotic fracture were FRAX[®]: 71.4% (95% CI: 70.6 to 72.2%), QFracture[®]: 61.4% (95% CI: 60.5 to 62.3%) and for hip fracture both 76.1% (95% CI: 74.9 to 77.2%). Prevalence of coded osteoporosis up to the index date was greater for patients (5.7%) compared to non-COPD subjects (3.9%), $p < 0.001$. In those without former osteoporosis, patients (n=73,084) had an increased osteoporosis incidence compared to non-COPD subjects (n=264,544), (adjusted hazard ratio, 1.13, 95% CI 1.05 to 1.22).

Conclusion COPD patients are at increased risk of fractures and osteoporosis. Despite this, there is no systematic assessment of fracture risk in clinical practice. Fracture risk tools identify those at high-risk of fracture in patients with COPD.

For peer review only

Strengths and limitations of this study

- This study examined electronic health records from a large, nationally representative sample of the UK population.
- A wide range of potential confounders were evaluated and adjusted for in the analyses.
- For the assessment of the fracture prediction tools, the population of patients with COPD used was large, with many fracture events, and included both men and women.
- Data collected in Read codes, in primary care represent only a snap-shot from a clinical consultation.
- The incidence of osteoporosis based on clinical codes, may reflect an underestimation of the true risk of osteoporosis since bone mineral density is not systematically assessed.

INTRODUCTION

Osteoporosis in both male and female patients with COPD is firmly established as one of the core comorbid conditions.[1,2] Over the last decade, it has become clear that osteoporosis is not just an end-stage COPD problem[3] nor just in those on maintenance oral corticosteroids, but it also occurs in a large proportion of those with mild-moderate airflow obstruction and even in steroid naïve patients.[4,5] The Global Initiative for Chronic Obstructive Pulmonary Disease (GOLD strategy recommends that osteoporosis co-existence should be considered in COPD [1] and the UK National Institute for Health and Care Excellence (NICE) Guidelines on osteoporosis considers COPD as a secondary cause of osteoporosis encouraging use of fracture prediction tools.[6]

The causes for osteoporosis in COPD are likely multiple and cumulative, including age, smoking exposure, inactivity, low body mass index (BMI), systemic inflammation and the frequent use of oral corticosteroids.[7] The clinical implications of osteoporosis include increased risk of fractures, poor quality of life, pain and further deterioration in lung function.[8,9] Osteoporosis can also remain undiagnosed as asymptomatic for many years.[10] Fractures are a function of trauma sustained, such as falls which are common in COPD [11], and the quality and architecture of bone. Fractures contribute further pain, poor quality of life, increased mortality and confer a substantial economic burden on health systems, patients and their families.[12,13] Given this, the individual risk of a future fracture in patients with COPD is crucial to determine in patient care and to treat accordingly.

Fracture risk prediction tools based on clinical and personal characteristics have been developed over the years to guide investigation and management of those identified to be at high risk of osteoporotic fractures, worldwide. These include for the UK (and many other regions), FRAX[®] and QFracture[®]. [6]

The full extent of fracture risk assessment in patients with COPD is not fully established. The aim of this study was to assess incidence of hip fracture or major osteoporotic fractures (MOF) in patients with COPD compared to non-COPD subjects and to evaluate the use and performance of fracture risk prediction tools in patients. Further, to assess the prevalence of coded osteoporosis up to the time of COPD diagnosis.

METHODS

Information for this cohort study was obtained from The Health Improvement Network (THIN), an anonymised primary care database representing 6.2% of the total UK population.[14]

The study population consisted of patients 40 years and over with a new Read coded COPD diagnosis during the data collection period 1/1/04-31/12/15, with at least 1 year of record prior to COPD diagnosis.[15] Each patient was matched by age, gender and GP practice to up to four subjects without a history of COPD to generate a matched cohort and assigned the same index date.

Follow up was from the index date to the first record of either the occurrence of the outcome of interest (fracture/osteoporosis), the date of transfer of the patient out of the practice area, death or end of THIN data collection. Read coded diagnoses for osteoporosis (*Appendix 1*) and Read coded hip fracture or major osteoporotic fractures (MOF) (fracture of the hip, proximal humerus, forearm or clinically symptomatic vertebra/spine) were ascertained.

A series of explanatory variables [6,16] determined at baseline (prior to or at index date) included: Charlson Comorbidity Index (CCI) score,[17] Townsend social deprivation score, fall, prior fractures, parental history of fall/osteoporosis, relevant comorbidities and secondary causes of osteoporosis as defined in the FRAX® questionnaire.[18] Records for smoking status, alcohol use, MRC Dyspnoea scale, BMI, and use of specific prescription drugs were restricted to a defined time period.

Individual follow-up time was divided into periods during which participants were considered exposed, or not exposed, to oral corticosteroids (a binary measure). Exposed periods started from prescription date until the first gap of more than 90 days between prescriptions; with individuals considered unexposed from the 91st day onwards. Individuals were considered exposed at study entry if they had received a relevant prescription within 90 days prior. The effect of exposure was assumed to be

constant, and not cumulative, over time (i.e. no time-dependent terms were entered
into the model).

Input variables included clinical status, prescription drug use, and demographic
characteristics, according to the variables/definitions used in both FRAX[®] and
QFracture[®] tools,[18,19], additional detail on the method is provided in an online data
supplement (*Appendix 2*). Imputation was used for missing variables.

The 10-year risk score for hip fracture and MOF according to QFracture[®] (version
2017.0.0.0) and FRAX[®] for UK without BMD information (desktop version 3.12) were
calculated for patients with COPD, aged 40-90 years old. A complete case sensitivity
analysis without imputed variables was also performed (*Appendix 3*).

**Statistical analyses**

Incidence rates were calculated for both groups using time-dependent Cox proportional
hazards regression to estimate hazard ratios (HRs) of osteoporosis and fracture risks,
with OCS treated as a time-dependent variable. Confounders were included in the final
model when independently changing the HRs for osteoporosis/fracture by at least 5%. A
former osteoporosis diagnosis or antiresorptive treatment prior to COPD diagnosis
excluded that subject from analyses related to either osteoporosis incidence or risk
(*Appendix 4*). In addition to evaluating incidence in the whole cohort, separate sub-
analyses excluded a) patients with COPD and no documented smoking history together
with their matched non-COPD subjects and b) those with no prior record of
osteoporosis.

To evaluate FRAX[®] and QFracture[®], the outcome was treated as a binary variable
(fracture or no fracture). Fracture risk probabilities were categorised based on
recommended treatment thresholds ($\geq 20\%$ for MOF and $\geq 3\%$ for hip fracture).[20] To
evaluate the overall ability of each tool to discriminate (performance) between those at
low and high risks, the area under the receiver operating characteristic (ROC) curve was
calculated. Sensitivity, specificity, positive and negative predictive values were
calculated. Survival analysis was performed and Kaplan-Meier plots comparing the
fracture incidence were generated.

All statistical analyses were performed using Stata 15.0 (StataCorp LP).

Patient involvement

The results and implications of previous research from the team on systematic assessment of osteoporosis in patients with COPD [4] has been discussed extensively in previous patient meetings. Whilst this and other literature has strengthened the GOLD strategy recommendations,[1] evaluation of clinical services would suggest systematic assessment is not done in patients. More recently, patients with COPD out-patient clinics have approached the principal investigator at the time of their “ad hoc osteoporosis” diagnosis to ask why this was not investigated at or closer to COPD diagnosis and how osteoporosis could be assessed. This has led to the development of this grant application with significant patient input in the design and context. The results have been discussed back with representatives on the respiratory research panel. Given the implications for clinical practice, the findings have been discussed extensively at the PPI meeting and a Breathe Easy meeting in early 2018. A lay summary has been developed for the patient newsletter (n>700) and website. In the meantime, members of the respiratory research panel are assisting the PI in planning future work regarding implementation.

RESULTS

The baseline characteristics are shown in Table 1. A total of 80,874 eligible patients with COPD and 308,999 matched non-COPD subjects were identified. The median follow-up period was 5 years for both patients and non-COPD subjects.

Osteoporosis at index date and incidence

Prevalence of coded osteoporosis up to the index date was greater for patients (5.7%) compared to non-COPD subjects (3.9%), $p < 0.001$. Within 1 year (before and after) of the index date, 1,504 (1.86%) patients with COPD had a new recorded diagnosis of osteoporosis compared to 3,059 (1.12%) in matched non-COPD subjects, $p < 0.001$. 3,186 (3.94%) of patients with COPD had a diagnosis of osteoporosis more than a year prior to index date compared to 8,822 (2.86%) for the matched controls $p < 0.001$. 1,457 (1.80%) patients with COPD compared to 3,694 (1.20%) non-COPD subjects, had a record of any diagnostic assessment for osteoporosis, recorded within 1 year (before and after) of the index date, ($p < 0.001$).

Demographics remained similar after excluding those with former osteoporosis. Patients with COPD ($n = 73,084$) compared to non-COPD subjects ($n = 264,544$) were significantly more likely to have incident diagnosis of osteoporosis (crude hazard ratio (HR), 1.96; 95% confidence interval [CI] 1.87 to 2.05; *Appendix 5*).

Incidence of Fracture

There was a significantly increased risk of both MOF, crude hazard ratio of 1.60 (95% CI 1.52 to 1.69) and hip fractures: 1.67 (95% CI 1.56 to 1.80) in patients with COPD compared to non-COPD subjects in the unadjusted model. In the fully adjusted models the association were diminished (Table 2). Smoking status altered the effect between COPD and fracture the most, followed by BMI, CCI score and oral corticosteroid.

Sensitivity analysis with participants with no former osteoporosis showed similar results. The risk of MOF was also similar when evaluated in only patients with COPD with a prior history of smoking and their matched controls. However, here, the risk of hip fracture remained significantly increased in the adjusted model compared to non-COPD subjects (aHR, 1.13; 95% CI 1.004 to 1.280; p -value: 0.043).

Fracture risk prediction tools in COPD

Only 1074 (1.33%) of patients with COPD had a FRAX[®] assessment READ coded ever documented in the records and 12 patients had a READ coded QFracture[®] assessment. Within 1 year (before and after) of index date, 248 (0.31%) of patients with COPD had a FRAX[®] and only 1 patient a QFracture[®].

The final population for the discriminatory accuracy analysis comprised 72,559 patients aged 40-90 years with COPD and no prior diagnosis of osteoporosis or prescription of any anti-resorptive treatment (*demographics in Appendix 6*). This included 4,605 (6.4%) who experienced a MOF and 1,444 (2.0%) who experienced hip fracture.

When the FRAX[®] and QFracture[®] scores were calculated for patients with COPD, for hip fracture 29,035 (40.0%) had a risk $\geq 3\%$ using FRAX[®] and 33,065 (45.6%) using the QFracture[®]. For MOF, 6,221 (8.6%) of patients had a risk $\geq 20\%$ using FRAX[®] and 9,546 (13.2%) using QFracture[®].

Both risk tools had a similar discriminatory accuracy for hip fracture (FRAX[®] 76.1%, 95% CI 74.9 to 77.2% and QFracture[®] 76.1%, 95% CI 74.9 to 77.2%). FRAX[®], however, had a higher accuracy for MOF (71.4% 95% CI 70.6% to 72.2%) than QFracture[®] (61.4% 95% CI 60.5% to 62.3%).

The discriminatory accuracies were better in women than men. The performance of the prediction tools was similar in the patients aged 50-90 years compared to 40-90-year olds. Table 3 shows the results for the sensitivity, specificity, positive and negative predictive values assessed for the performance of the prediction tools at $\geq 3\%$ risk probability for hip fracture and $\geq 20\%$ risk probability for major osteoporotic fractures. At a 20% fracture risk cut-off for MOF, FRAX[®] identified 25.4% (95% CI, 22.7% to 28.1%) (sensitivity) of those who went on to experience an MOF, QFracture[®] was 25.2% (95% CI, 22.5% to 27.9%). The specificity, positive predictive value (PPV) and negative predictive value (NPV) were 92.6% (95% CI, 91.0 to 94.2), 18.8% (95% CI, 16.4% to 21.1%) and 94.8% (95% CI, 93.4% to 96.2%) for FRAX[®] and 87.7% (95% CI, 85.7% to 89.7%), 12.2% (95% CI, 10.2% to 14.2%) and 94.5% (95% CI, 93.1% to 95.9%) respectively for QFracture[®]. At a 3% risk cut-off for hip fractures, FRAX[®] sensitivity, specificity, PPV and NPV were 78.1% (95% CI, 75.6% to 80.7%), 60.8%

(95% CI, 57.8% to 63.8%), 3.9% (95% CI, 2.7% to 5.1%), 99.3% (95% CI, 98.8% to
99.8%) respectively and 82.1% (95% CI, 79.7% to 84.5%), 55.2% (95% CI, 52.1% to
58.3%), 3.6% (95% CI, 2.5% to 4.8%) and 99.3% (95% CI, 98.8% to 99.8%)
respectively for QFracture®.

The Kaplan-Meier plots for time to first MOF for QFracture® and FRAX® are presented in
Figure 1.

DISCUSSION

Using UK primary care electronic health records, we have reported on the burden of fractures in patients with COPD with both hip and MOF increased in patients with COPD compared to age, gender and GP surgery matched subjects. Despite the increased fracture risk and recommendations in the NICE osteoporosis guidelines, fracture risk prediction tools are rarely coded. However, where the risk score was retrospectively calculated, the risk prediction tools identify those at risk of hip fracture or MOF. Therefore, fracture risk prediction and subsequent targeted therapy and management could transform multi-morbidity management of COPD. In addition, we report that the prevalence and incidence of osteoporosis, a risk for fracture, in patients with COPD, is far greater than in non-COPD subjects.

Prevalence of osteoporosis varies widely in the different studies of patients with COPD. This is mainly dependent on the severity of COPD,[4,5] whether osteoporosis was systematically sought or self-reported [4,21], and whether patients included were on oral corticosteroids.[3] A prevalence of 23-32% has been reported where BMD was systematically performed [22].[4], while 14% of patients with COPD self-reported osteoporosis compared to 5% in those without COPD.[21] The prevalence of coded osteoporosis in the GP health records was, however, far lower at 5.7% than the reported prevalence from clinical studies when osteoporosis and BMD are systematically assessed. This raises the question of subclinical, undiagnosed disease leading to a missed opportunity for intervention and strengthening the need for a systematic assessment especially when cost-efficient anti-resorptive treatment is available.[23]

[revised manuscript text omitted]

addressing prevention and treatment of those at greatest risk of fracture has the potential to improve outcomes for patients with COPD.

For peer review only

Acknowledgements

With grateful thanks to Prof J Hippisley-Cox and Mr S Hippisley-Cox for use of the QFracture®.

Competing interests

All authors have completed the Unified Competing Interest form (available on request from the corresponding author) and declare: CEB, TMM, JES received an investigator sponsored study grant from Pfizer for the submitted work; CEB reports grants from MRC/Association of British Pharmaceutical Industry (ABPI), TSB, GSK and other support from Chiesi and Boehringer, outside the submitted work; JES reports personal fees from Astra Zeneca, Boehringer-Ingelheim, Nutricia, Chiesi, Sandoz, Novartis, Pfizer, MIMS, RCGP, Cogora and other support from PCRS-UK, Education for Health, Teva and NICE outside the submitted work; no financial relationships with any organisations that might have an interest in the submitted work in the previous three years, no other relationships or activities that could appear to have influenced the submitted work.

Author contributions

CEB, TMM and JES designed study concept and design and are grant holders. RKA conducted the main statistical analysis and wrote the first draft of the manuscript. JEG prepared the THIN data extracts used and assisted with analysis. All authors contributed to the interpretation of the data, writing of the manuscript and critical revisions.

CEB is guarantor.

Ethical approval

The study was approved by an independent Scientific Review Committee (SRC), 16THIN029.

Funding

This study was funded by a COPD "Open Air" research grant from Pfizer.

The funders had no role in study design, data collection and analysis, decision to publish or preparation of the manuscript. CEB and TMM are supported by the NIHR Nottingham BRC. The views expressed are those of the authors and not necessarily those of the NHS, the NIHR or the Department of Health.

Data sharing

No additional data available.

For peer review only

REFERENCES

[revised manuscript text omitted]

*Validation, and Updating*. Springer 2008. doi:10.1080/10543400903244270

- Kanis JA, Oden A, Johnell O, *et al*. The use of clinical risk factors enhances the
performance of BMD in the prediction of hip and osteoporotic fractures in men and
women. *Osteoporos Int* 2007;**18**:1033–46. doi:10.1007/s00198-007-0343-y
Hippisley-Cox J, Coupland C. Derivation and validation of updated QFracture algorithm
to predict risk of osteoporotic fracture in primary care in the United Kingdom:
prospective open cohort study. *BMJ* 2012;**344**:e3427. doi:10.1136/bmj.e3427
Cummings SR, Melton LJ. Epidemiology and outcomes of osteoporotic fractures.
*Lancet* 2002;**359**:1761–7. doi:10.1016/S0140-6736(02)08657-9
National Institute for Health and Care Excellence, NICE. Chronic obstructive
pulmonary disease in over 16s: diagnosis and management. NICE 2010.
British Thoracic Society, Pulmonary Rehabilitation Guideline, Group. BTS Guideline on
Pulmonary Rehabilitation in Adults. *Thorax* 2013;**68**.

Table 1: Baseline characteristics of patients with COPD and non-COPD subjects

Descriptor	COPD patients		Non-COPD subjects		p-value
	n = 80,874	%	n = 308,999	%	
Mean age at index date (years, SD)	66.9 (11.0)		66.5 (10.9)		
Gender					0.002
Male	42,799	52.9	161,648	52.3	
Female	38,075	47.1	147,351	47.7	
Follow-up (years, median, IQR)	5.28	2.6-8.3	5.24	2.6-8.3	
MRC Dyspnoea Scale (1 Year either side of diagnosis)					<0.001
1	9,499	11.8	1,168	0.4	
2	19,466	24.1	1,092	0.4	
3	10,488	13.0	446	0.1	
4 & 5	5,237	6.5	177	0.1	
No record	36,184	44.7	306,116	99.1	
Charlson Comorbidity Index Score					<0.001
0	0	0.0	172,566	55.9	
1	41,777	51.7	50,955	16.5	
2	13,506	16.7	42,667	13.8	
3	12,694	15.7	23,546	7.6	
≥ 4	12,897	16.0	19,265	6.2	
Body Mass Index (BMI) (kg/m²)					<0.001
Underweight (< 18.5)	3,414	4.2	2,699	0.9	
Normal (18.5 – 24.9)	24,734	30.6	54,267	17.6	
Overweight (25 – 29.9)	23,497	29.1	77,129	25.0	
Obese (≥30)	19,083	23.6	60,280	19.5	
No BMI	10,146	12.6	114,624	37.1	
Smoking status (1 Year either side of diagnosis)					<0.001
Never smoked	7,925	9.8	94,800	30.7	
Ex-smoker	38,590	47.7	72,989	23.6	
Current smoker	32,436	40.1	34,691	11.2	
Unknown	1,923	2.4	106,519	34.5	
History of Falls (prior to or at diagnosis)					
Personal history	8,969	11.1	26,203	8.5	<0.001
Parental history of fall/osteoporosis	96	0.1	298	0.1	0.076
Medications (1 Year either side of diagnosis)					
Oral Glucocorticoid Use	33,618	41.6	19,479	6.3	<0.001
Inhaled Corticosteroid Use	47,574	58.8	21,312	6.9	<0.001

For peer review only

Table 2: Risk of fractures in patients with COPD compared with non-COPD subjects

	Number of fractures	Rate/1,000 person-years	Crude HR (95% CI)	Fully adjusted HR (95% CI)
Major osteoporotic fractures				
Non-COPD subjects	6,032	4.32 (4.22 – 4.44)	Reference	Reference
Patients with COPD	2,234	6.64 (6.37 – 6.92)	1.60 (1.52 – 1.69)	1.04 (0.96 – 1.12) ^a
Hip fracture				
Non-COPD subjects	3,170	2.26 (2.18 – 2.34)	Reference	Reference
Patients with COPD	1,213	3.57 (3.38 – 3.78)	1.67 (1.56 – 1.80)	1.09 (0.98 – 1.21) ^b

HR – Hazard ratio; CI – Confidence interval

Crude HR – Cox regression model derived HR adjusted for age, sex, and GP practice

^a Multivariate Cox regression model derived HR was adjusted for age, sex, GP practice, Charlson Comorbidity Index, Body Mass Index, smoking status, inhaled corticosteroid use, antidepressant use and cumulative oral corticosteroid use.

^b Multivariate Cox regression model derived HR was adjusted for age, sex, GP practice, Charlson Comorbidity Index, Body Mass Index, smoking status, inhaled corticosteroid use and cumulative oral corticosteroid use.

Table 3: Discrimination measures for FRAX® and QFracture® at recommended treatment cut offs for both major osteoporotic and hip fractures

Discriminatory measures	FRAX®	QFracture®
	Measure for ≥ 20% risk (95% CI)	Measure for ≥ 20% risk (95% CI)
Major Osteoporotic fractures		
Sensitivity	25.4% (22.7-28.1%)	25.2% (22.5-27.9%)
Specificity	92.6% (91.0-94.2%)	87.7% (85.7-89.7%)
Positive Predictive Value	18.8% (16.4-21.1%)	12.2% (10.2-14.2%)
Negative Predictive Value	94.8% (93.4-96.2%)	94.5% (93.1-95.9%)
	Measure for ≥ 3% risk	Measure for ≥ 3% risk
Hip fracture		
Sensitivity	78.1% (75.6-80.7%)	82.1% (79.7-84.5%)
Specificity	60.8% (57.8-63.8%)	55.2% (52.1-58.3%)
Positive Predictive Value	3.9% (2.7-5.1%)	3.6% (2.5-4.8%)
Negative Predictive Value	99.3% (98.8-99.8%)	99.3% (98.8-99.8%)

CI – Confidence interval

Figure 1: Kaplan-Meier plots comparing incidence of major osteoporotic fractures at various predicted fracture risk categories in patients with COPD using (a) FRAX® and (b) QFracture®

For peer review only

(a)

(b)

Appendix 1

Read code definitions for selected input variables

Variable	Read codes
COPD	H3...00, H3...11, H31..00, H310.00, H310000, H310z00, H311.00, H311000, H311100, H311z00, H312.00, H312000, H312011, H312100, H312300, H312z00, H313.00, H31y.00, H31y100, H31yz00, H31z.00, H32..00, H320.00, H320000, H320100, H320200, H320300, H320311, H320z00, H321.00, H322.00, H32y.00, H32y000, H32y100, H32y111, H32y200, H32yz00, H32z.00, H36..00, H37..00, H38..00, H39..00, H3A..00, H3y..00, H3y..11, H3z..00, H3z..11
Osteoporosis	5850.00, 58E4.00, 58E8.00, 58EA.00, 58EE.00, 58EG.00, 58EK.00, 58EM.00, 58ES.00, 58EV.00, 7230A, 7230B, 7230D, 7230PM, 7230PT, N330.00, N330000, N330100, N330200, N330300, N330400, N330500, N330600, N330700, N330800, N330900, N330A00, N330B00, N330C00, N330D00, N330z00, N331200, N331300, N331400, N331500, N331600, N331800, N331900, N331A00, N331B00, N331M00, N331N00, NyuB000, NyuB100, NyuB200, NyuB800
Antiresorptive treatment (drug code)	97138998, 99158998, 99158997, 97139998, 96920998, 96789998, 93478998, 97140998, 97218998, 93975992, 83457998, 97064992, 83456998, 96897998, 96020992, 96901998, 95879992, 98249990, 97031992, 98581990, 99018990, 98198990, 62945979, 96737998, 97066992, 97051992, 97780990, 98199990, 61594979, 99261990, 96604992, 92004979, 97248990, 99263990, 94089992, 93127992, 94756992, 91526998, 89828998, 88144998, 88144997, 88225998, 89434998, 93502998, 99862998, 95304998, 93228997, 96904998, 93228998, 95304996, 99862997, 95304997, 93228996, 99864998, 91997998, 91998998, 87933998, 81073998, 61612979, 87155998, 87154998, 88542998, 91378998, 82066998,

Antiresorptive treatment (drug code)	82065998, 81256998, 81255998, 91190996, 89518998, 91190998, 91191998, 86599998, 91190997, 91191997, 93692990, 81472998, 94276990, 93827990, 92431990, 94161990, 93610990, 94245990, 61524979, 99883979, 93828990, 99867979, 95572998, 99758998, 96764998, 97398992, 95244990, 89367998, 86562998, 86561998, 87645998, 87644998, 86079998, 86076998, 91533998, 87151998, 81270998, 91027998, 93617996, 93618996, 93618997, 93617997, 90527998, 86566998, 91028998, 87137998, 87136998, 91674998, 86564998, 86567998, 87135998, 93089979, 99357998, 84212998, 84691998, 89021998, 91764998, 90551998, 91763998, 81869998, 91764997, 91763997, 89354979, 92813997, 93402998, 92813998, 98527996, 93403996, 98527998, 93403998, 93402996, 84531998, 58602979, 87606998, 85936998, 81112998, 97865998, 85935998, 81111998, 76983978, 83078978
--	--

Appendix 2

METHODS

Potential confounders

For smoking status, alcohol use, MRC Dyspnoea scale, and a list of prescription drugs, the most recent record within 1 year (before and after) of index date were used. A BMI record within 2 years (before and after) of index date was used.

Where possible BMI was calculated from height and weight records, for patients with a missing BMI record. The BMI was subsequently categorised (underweight: <18.5 kg/m², normal: 18.5-<25 kg/m², overweight: 25-<30 kg/m², obese: >30 kg/m²).

Having received at least one prescription for inhaled corticosteroids, anti-epileptics, antidepressants, oestrogen-only Hormone Replacement Therapy (HRT) and osteoporosis medications, within 1 year (before and after) of index date were considered as risk factors.

Prediction tools – Input variables

The respective variable definitions as outlined in the algorithms for the prediction tools were used.

Smoking status – In QFracture[®], three current smoking categories are provided according to the number of cigarettes smoked daily[1]. To avoid the bias of categorising patients in one of the outlying categories, “current smokers” with no documented number of cigarettes smoked were assigned to the middle category “10-19 cigarettes daily” as done in a recent publication [2]. For FRAX[®]’s two-category smoking status, former smokers were assigned to the “non-smoker” category as was done in the cohorts used to develop FRAX[®]. [3]

Alcohol consumption – similarly, for alcohol use in QFracture[®], alcohol drinkers with no documented unit/day intake were assigned to “moderate (3-6units/day)”.

Missing values for BMI, smoking status, and alcohol use were imputed by multiple imputation using all predictors, resulting in twenty imputed datasets.[4] A complete case sensitivity analysis without imputed variables was also performed (Appendix 3).

References

- 1 ClinRisk Ltd. QFracture-2016® risk calculator. <http://www.qfracture.org/> (accessed 20 Sep 2017).
- 2 Dagan N, Cohen-Stavi C, Leventer-Roberts M, *et al.* External validation and comparison of three prediction tools for risk of osteoporotic fractures using data from population based electronic health records: retrospective cohort study. *BMJ* 2017;**356**:i6755. doi:10.1136/BMJ.I6755
- 3 Kanis JA, Oden A, Johnell O, *et al.* The use of clinical risk factors enhances the performance of BMD in the prediction of hip and osteoporotic fractures in men and women. *Osteoporos Int* 2007;**18**:1033–46. doi:10.1007/s00198-007-0343-y
- 4 Horton NJ, Lipsitz SR. Multiple Imputation in Practice. *Am Stat* 2001;**55**:244–54. doi:10.1198/000313001317098266

Appendix 3

Fracture risk prediction tools in COPD (Complete case analysis)

Of the 72,559 patients aged 40-90 years with COPD and no prior diagnosis of osteoporosis or prescription of any anti-resorptive treatment, 41,879 (57.7%) of patients had complete data. Amongst the patients with complete data, 2,649 (6.3%) experienced a MOF and 806 (1.9%) experienced hip fracture.

Both risk tools had about the same discriminatory accuracy as that obtained from the entire cohort with imputed data. The AUC for hip fracture was 75.6%, 95% CI 74.0% to 77.1% for FRAX® and 75.6%, 95% CI 74.0% to 77.2% for QFracture®. FRAX® maintained a higher accuracy for MOF (71.6%, 95% CI 70.6% to 72.6%) than QFracture® (61.1%, 95% CI 60.0% to 62.2%).

Appendix 4

Figure E1: Study population flow diagram

Appendix 5

Table E1: Risk of osteoporosis in patients with COPD compared with non-COPD subjects

Descriptor	Crude HR (95% CI)	Fully adjusted HR (95% CI)
COPD		
Non-COPD subjects	Reference	Reference
COPD patients	1.96 (1.87 – 2.06)	1.13 (1.05 – 1.22)
Charlson Comorbidity Index		
Score 0	Reference	Reference
Score 1	1.27 (1.18 – 1.36)	1.14 (1.06 – 1.23)
Score 2	1.34 (1.24 – 1.44)	1.27 (1.17 – 1.37)
Score 3	1.41 (1.28 – 1.55)	1.29 (1.17 – 1.42)
Score 4 & more	1.48 (1.33 – 1.64)	1.44 (1.29 – 1.61)
Body Mass Index (kg/m²)		
Underweight (<18.5)	1.93 (1.64 – 2.27)	1.91 (1.63 – 2.25)
Normal (18.5 – 24.9)	Reference	Reference
Overweight (25 – 29.9)	0.64 (0.60 – 0.69)	0.63 (0.58 – 0.67)
Obese (≥ 30)	0.47 (0.43 – 0.51)	0.45 (0.41 – 0.48)
No record	0.50 (0.46 – 0.53)	0.57 (0.52 – 0.61)
Smoking status		
Never	Reference	Reference
Ex	1.01 (0.95 – 1.08)	1.02 (0.95 – 1.09)
Current	1.23 (1.13 – 1.33)	1.15 (1.06 – 1.25)
Unknown	0.69 (0.64 – 0.74)	0.77 (0.71 – 0.83)
Oral Corticosteroid Use		
Unexposed	Reference	Reference
Exposed	2.79 (2.56 – 3.05)	1.91 (1.73 – 2.10)
Inhaled Corticosteroid Use		
No	Reference	Reference
Yes	1.35 (1.26 – 1.45)	1.24 (1.15 – 1.34)

HR – Hazard ratio; CI – Confidence interval

Crude HR – Cox regression model derived HR adjusted for age, sex, and GP practice

The adjusted Hazard Ratio (aHR) was 1.13, 95% CI 1.05 to 1.22, $p < 0.0001$ – the multivariate Cox regression model derived aHR was adjusted for age, sex, GP practice, Charlson comorbidity index, body mass index, smoking status, inhaled corticosteroid use, and cumulative oral corticosteroid use.

Appendix 6

Table E2: Baseline characteristics of patients with COPD aged 40-90 years with no prior diagnosis of osteoporosis or prescription of any anti-resorptive treatment

Descriptor	COPD patients	
	n = 72,559	%
Mean age at index date (years, SD)	66.1 (10.7)	
Gender		
Female	31,885	43.9
MRC Dyspnoea Scale (1 Year either side of diagnosis)		
1	8,882	12.2
2	17,718	24.4
3	9,257	12.8
4 & 5	4,346	6.0
No record	32,356	44.6
Charlson Comorbidity Index Score		
0	0	0
1	38,573	53.2
2	11,953	16.5
3	11,110	15.3
≥ 4	10,923	15.1
Body Mass Index (BMI) (kg/m²)		
Underweight (< 18.5)	2,730	3.8
Normal (18.5 – 24.9)	21,791	30.0
Overweight (25 – 29.9)	21,504	29.6
Obese (≥30)	17,627	24.3
No BMI	8,907	12.3
Smoking status (1 Year either side of diagnosis)		
Never smoked	7,062	9.7
Ex-smoker	33,810	46.6
Current smoker	29,949	41.3
Unknown	1,738	2.4

STROBE 2007 (v4) Statement—Checklist of items that should be included in reports of cohort studies

Section/Topic	Item #	Recommendation	Reported on page #
Title and abstract	1	(a) Indicate the study’s design with a commonly used term in the title or the abstract	1
		(b) Provide in the abstract an informative and balanced summary of what was done and what was found	2
Introduction			
Background/rationale	2	Explain the scientific background and rationale for the investigation being reported	5
Objectives	3	State specific objectives, including any pre-specified hypotheses	5
Methods			
Study design	4	Present key elements of study design early in the paper	6
Setting	5	Describe the setting, locations, and relevant dates, including periods of recruitment, exposure, follow-up, and data collection	6
Participants	6	(a) Give the eligibility criteria, and the sources and methods of selection of participants. Describe methods of follow-up	6
		(b) For matched studies, give matching criteria and number of exposed and unexposed	6
Variables	7	Clearly define all outcomes, exposures, predictors, potential confounders, and effect modifiers. Give diagnostic criteria, if applicable	6,7
Data sources/ measurement	8*	For each variable of interest, give sources of data and details of methods of assessment (measurement). Describe comparability of assessment methods if there is more than one group	6,7
Bias	9	Describe any efforts to address potential sources of bias	7
Study size	10	Explain how the study size was arrived at	-
Quantitative variables	11	Explain how quantitative variables were handled in the analyses. If applicable, describe which groupings were chosen and why	7
Statistical methods	12	(a) Describe all statistical methods, including those used to control for confounding	7
		(b) Describe any methods used to examine subgroups and interactions	-
		(c) Explain how missing data were addressed	6, Appendix 2
		(d) If applicable, explain how loss to follow-up was addressed	-
		(e) Describe any sensitivity analyses	7
Results			
Participants	13*	(a) Report numbers of individuals at each stage of study—eg numbers potentially eligible, examined for eligibility, confirmed	9

		eligible, included in the study, completing follow-up, and analysed	
		(b) Give reasons for non-participation at each stage	-
		(c) Consider use of a flow diagram	-
Descriptive data	14*	(a) Give characteristics of study participants (eg demographic, clinical, social) and information on exposures and potential confounders	9, Table 1 (17)
		(b) Indicate number of participants with missing data for each variable of interest	Table 1 (19)
		(c) Summarise follow-up time (eg, average and total amount)	9
Outcome data	15*	Report numbers of outcome events or summary measures over time	9, 10
Main results	16	(a) Give unadjusted estimates and, if applicable, confounder-adjusted estimates and their precision (eg, 95% confidence interval). Make clear which confounders were adjusted for and why they were included	9, 10, Table 2 (20), Appendix 5
		(b) Report category boundaries when continuous variables were categorized	Appendix 2
		(c) If relevant, consider translating estimates of relative risk into absolute risk for a meaningful time period	-
Other analyses	17	Report other analyses done—eg analyses of subgroups and interactions, and sensitivity analyses	9,10,11
Discussion			
Key results	18	Summarise key results with reference to study objectives	12
Limitations			
Interpretation	20	Give a cautious overall interpretation of results considering objectives, limitations, multiplicity of analyses, results from similar studies, and other relevant evidence	12-14
Generalisability	21	Discuss the generalisability (external validity) of the study results	14
Other information			
Funding	22	Give the source of funding and the role of the funders for the present study and, if applicable, for the original study on which the present article is based	15

*Give information separately for cases and controls in case-control studies and, if applicable, for exposed and unexposed groups in cohort and cross-sectional studies.

Note: An Explanation and Elaboration article discusses each checklist item and gives methodological background and published examples of transparent reporting. The STROBE checklist is best used in conjunction with this article (freely available on the Web sites of PLoS Medicine at <http://www.plosmedicine.org/>, Annals of Internal Medicine at <http://www.annals.org/>, and Epidemiology at <http://www.epidem.com/>). Information on the STROBE Initiative is available at www.strobe-statement.org.

BMJ Open

Predicting Fracture Risk in Patients with Chronic Obstructive Pulmonary Disease: A UK-based Population-based Cohort Study

Journal:	BMJ Open
Manuscript ID	bmjopen-2018-024951.R2
Article Type:	Research
Date Submitted by the Author:	23-Jan-2019
Complete List of Authors:	Akyea, Ralph; University of Nottingham, Nottingham Respiratory Research Unit, NIHR Nottingham Biomedical Research Centre; University of Nottingham, Division of Epidemiology & Public Health McKeever, Tricia; University of Nottingham, Nottingham Respiratory Research Unit, NIHR Nottingham Biomedical Research Centre; University of Nottingham, Division of Epidemiology & Public Health Gibson, Jack; University of Nottingham, Division of Epidemiology & Public Health Scullion, Jane; Institute for Lung Health, University Hospitals of Leicester Glenfield Site Bolton, Charlotte; University of Nottingham, Nottingham Respiratory Research Unit, NIHR Nottingham Biomedical Research Centre
Primary Subject Heading:	Respiratory medicine
Secondary Subject Heading:	Epidemiology
Keywords:	Fracture, COPD, fracture risk prediction tool, osteoporosis

**Predicting Fracture Risk in Patients with Chronic Obstructive**
**Pulmonary Disease: A UK-based Population-based Cohort Study**

Ralph K Akyea^{1, 2}; Tricia M McKeever^{1, 2}; Jack E Gibson²; Jane E Scullion³; Charlotte E
Bolton¹

**Authors affiliations:**

¹ Nottingham Respiratory Research Unit, NIHR Nottingham Biomedical Research
Centre, School of Medicine, University of Nottingham, UK.

² Division of Epidemiology and Public Health, School of Medicine, University of
Nottingham, UK.

³ University Hospitals of Leicester Glenfield Site, Institute for Lung Health, Leicester,
UK.

**Corresponding author information:**

Professor Charlotte E Bolton,

Nottingham Respiratory Research Unit, NIHR Nottingham Biomedical Research
Centre, School of Medicine, University of Nottingham, City Hospital Campus, Hucknall
Road, Nottingham, NG5 1PB, UK.

Tel: +44 (0)115 8231710

Email: charlotte.bolton@nottingham.ac.uk

**Keywords:**

Fracture, osteoporosis, COPD, fracture risk prediction tool

**Word counts:**

Abstract: 297

Main text: 3,024

1 **ABSTRACT**

[revised manuscript text omitted]

19 ($n = 264,544$) were significantly more likely to have incident diagnosis of osteoporosis
20 (hazard ratio (HR), 1.96; 95% confidence interval [CI] 1.87 to 2.05; *Appendix 5*).

Incidence of Fracture

There was a significantly increased risk of MOF, hazard ratio of 1.60 (95% CI 1.52 to
1.69) and hip fractures alone: 1.67 (95% CI 1.56 to 1.80) in patients with COPD
compared to non-COPD patients. In the fully adjusted models the associations were
diminished (Table 2). Smoking status altered the effect between COPD and fracture
the most, followed by BMI, CCI score and OCS.

Sensitivity analysis with participants with no former osteoporosis showed similar
results. The risk of MOF was also similar when evaluated in only patients with COPD
with a documented prior history of smoking and their matched controls. However,
here, the risk of hip fracture remained significantly increased in the adjusted model
compared to non-COPD patients (aHR, 1.13; 95% CI 1.004 to 1.280; p -value:
0.043).

Fracture risk prediction tools in COPD

Only 1074 (1.33%) of patients with COPD had a FRAX[®] assessment READ coded
ever documented in the records and 12 patients had a READ coded QFracture[®]
assessment. Within 1 year (before and after) of index date, 248 (0.31%) of patients
with COPD had a FRAX[®] and only 1 patient a QFracture[®].

The final population for the discriminatory accuracy analysis comprised 72,559
patients aged 40-90 years with COPD and no prior diagnosis of osteoporosis or
prescription of any anti-resorptive treatment (*demographics in Appendix 6*). This
included 4,605 (6.4%) patients who experienced any MOF and 1,444 (2.0%) who
experienced a hip fracture.

When the FRAX[®] and QFracture[®] scores were calculated for patients with COPD, for
hip fracture there were 29,035 (40.0%) patients who had a risk $\geq 3\%$ using FRAX[®]
and 33,065 (45.6%) patients using the QFracture[®]. For any MOF, 6,221 (8.6%) of

[revised manuscript text omitted]

4

For peer review only

1 **Acknowledgements**

With grateful thanks to Prof J Hippisley-Cox and Mr S Hippisley-Cox for use of the
QFracture®.

**Competing interests**

All authors have completed the Unified Competing Interest form (available on
request from the corresponding author) and declare: CEB, TMM, JES received an
investigator sponsored study grant from Pfizer for the submitted work; CEB reports
grants from MRC/Association of British Pharmaceutical Industry (ABPI), TSB, GSK
and other support from Chiesi and Boehringer, outside the submitted work; JES
reports personal fees from Astra Zeneca, Boehringer-Ingelheim, Nutricia, Chiesi,
Sandoz, Novartis, Pfizer, MIMS, RCGP, Cogora and other support from PCRS-UK,
Education for Health, Teva and NICE outside the submitted work; no financial
relationships with any organisations that might have an interest in the submitted
work in the previous three years, no other relationships or activities that could
appear to have influenced the submitted work.

**Author contributions**

CEB, TMM and JES designed study concept and design and are grant holders. RKA
conducted the main statistical analysis and wrote the first draft of the manuscript.
JEG prepared the THIN data extracts used and assisted with analysis. All authors
contributed to the interpretation of the data, writing of the manuscript and critical
revisions.

CEB is guarantor.

**Ethical approval**

The study was approved by an independent Scientific Review Committee (SRC),
16THIN029.

**Funding**

This study was funded by a COPD "Open Air" research grant from Pfizer.

The funders had no role in study design, data collection and analysis, decision to
publish or preparation of the manuscript. CEB and TMM are supported by the NIHR
Nottingham BRC. The views expressed are those of the authors and not necessarily
those of the NHS, the NIHR or the Department of Health.

**Data sharing**

No additional data available.

REFERENCES

- Global Strategy for the Diagnosis Management and Prevention of COPD. Global
Initiative for Chronic Obstructive Lung Disease (GOLD) 2017.
[http://goldcopd.org/gold-2017-global-strategy-diagnosis-management-prevention-](http://goldcopd.org/gold-2017-global-strategy-diagnosis-management-prevention-copd/)
[copd/](http://goldcopd.org/gold-2017-global-strategy-diagnosis-management-prevention-copd/) (accessed 13 Apr 2017).
- Barnes PJ, Celli BR. Systemic manifestations and comorbidities of COPD. *Eur Respir*
*J* 2009;**33**:1165–85. doi:10.1183/09031936.00128008
- Shane E, Silverberg SJ, Donovan D, *et al.* Osteoporosis in lung transplantation
candidates with end-stage pulmonary disease. *Am J Med* 1996;**101**:262–9.
- Bolton CE, Ionescu AA, Shiels KM, *et al.* Associated Loss of Fat-free Mass and Bone
Mineral Density in Chronic Obstructive Pulmonary Disease. *Am J Respir Crit Care*
*Med* 2004;**170**:1286–93. doi:10.1164/rccm.200406-754OC
- Duckers JM, Evans BA, Fraser WD, *et al.* Low bone mineral density in men with
chronic obstructive pulmonary disease. *Respir Res* 2011;**12**:101.
doi:10.1186/1465-9921-12-101
- National Institute for Health and Care Excellence. Osteoporosis: assessing the risk
of fragility fracture. 2012.
- Lehouck A, Boonen S, Decramer M, *et al.* COPD, Bone Metabolism, and
Osteoporosis. *Chest* 2011;**139**:648–57. doi:10.1378/chest.10-1427
- Gjertsen J-E, Baste V, Fevang JM, *et al.* Quality of life following hip fractures:
results from the Norwegian hip fracture register. *BMC Musculoskelet Disord*
2016;**17**. doi:10.1186/s12891-016-1111-y
- Jameson JL, De Groot LJ. *Endocrinology: Adult and Pediatric E-Book*. Elsevier
Health Sciences 2015.
- Peppas NA, Hilt JZ, Thomas JB. *Nanotechnology in Therapeutics: Current*
*Technology and Applications*. Horizon Bioscience 2007.
- Hakamy A, Bolton CE, Gibson JE, *et al.* Risk of fall in patients with COPD. *Thorax*
2018;**73**:1079–80. doi:10.1136/thoraxjnl-2017-211008
- Coughlan T, Dockery F. Osteoporosis and fracture risk in older people. *Clin Med*
2014;**14**:187–91. doi:10.7861/clinmedicine.14-2-187
- Roux C, Wyman A, Hooven FH, *et al.* Burden of non-hip, non-vertebral fractures on
quality of life in postmenopausal women. *Osteoporos Int* 2012;**23**:2863–71.
doi:10.1007/s00198-012-1935-8
- Blak BT, Thompson M, Dattani H, *et al.* Generalisability of The Health Improvement
Network (THIN) database: demographics, chronic disease prevalence and mortality
rates. *Inform Prim Care* 2011;**19**:251–5.
- Quint JK, Müllerova H, DiSantostefano RL, *et al.* Validation of chronic obstructive
pulmonary disease recording in the Clinical Practice Research Datalink (CPRD-
GOLD). *BMJ Open* 2014;**4**:e005540. doi:10.1136/bmjopen-2014-005540
- National Health Service. Read Codes.
2017.<https://digital.nhs.uk/services/terminology-and-classifications/read-codes>
(accessed 29 Dec 2018).
- Scottish Intercollegiate Guidelines Network (SIGN). Management of osteoporosis
and the prevention of fragility fractures. Edinburgh: 2015.
- Lix LM, Quail J, Teare G, *et al.* Performance of comorbidity measures for predicting

[revised manuscript text omitted]

Table 1: Baseline characteristics of patients with COPD and non-COPD patients

Descriptor	COPD patients		Non-COPD patients		p-value
	n = 80,874	%	n = 308,999	%	
Mean age at index date (years, SD)	66.9 (11.0)		66.5 (10.9)		
Sex					0.002
Male	42,799	52.9	161,648	52.3	
Female	38,075	47.1	147,351	47.7	
Follow-up (years, median, IQR)	5.28	2.6-8.3	5.24	2.6-8.3	
MRC Dyspnoea Scale (1 Year either side of diagnosis)					<0.001
1	9,499	11.8	1,168	0.4	
2	19,466	24.1	1,092	0.4	
3	10,488	13.0	446	0.1	
4 & 5	5,237	6.5	177	0.1	
No record	36,184	44.7	306,116	99.1	
Charlson Comorbidity Index Score					<0.001
0	0	0.0	172,566	55.9	
1	41,777	51.7	50,955	16.5	
2	13,506	16.7	42,667	13.8	
3	12,694	15.7	23,546	7.6	
≥ 4	12,897	16.0	19,265	6.2	
Body Mass Index (BMI) (kg/m²)					<0.001
Underweight (< 18.5)	3,414	4.2	2,699	0.9	
Normal (18.5 – 24.9)	24,734	30.6	54,267	17.6	
Overweight (25 – 29.9)	23,497	29.1	77,129	25.0	
Obese (≥30)	19,083	23.6	60,280	19.5	
No BMI	10,146	12.6	114,624	37.1	
Smoking status (1 Year either side of diagnosis)					<0.001
Never smoked	7,925	9.8	94,800	30.7	
Ex-smoker	38,590	47.7	72,989	23.6	
Current smoker	32,436	40.1	34,691	11.2	
Unknown	1,923	2.4	106,519	34.5	
History of Falls (prior to or at diagnosis)					
Personal history	8,969	11.1	26,203	8.5	<0.001
Parental history of fall/osteoporosis	96	0.1	298	0.1	0.076
Medications (1 Year either side of diagnosis)					
Oral corticosteroid Use (OCS)	33,618	41.6	19,479	6.3	<0.001
Inhaled Corticosteroid Use	47,574	58.8	21,312	6.9	<0.001

3

4

Table 2: Risk of all major osteoporotic fractures (MOF) and hip fractures alone in patients with COPD compared with non-COPD patients

	Number of fractures	Rate/1,000 person-years	HR (95% CI)	Fully adjusted HR (95% CI)
Major osteoporotic fractures (MOF)				
Non-COPD patients	6,032	4.32 (4.22 – 4.44)	Reference	Reference
Patients with COPD	2,234	6.64 (6.37 – 6.92)	1.60 (1.52 – 1.69)	1.04 (0.96 – 1.12) ^a
Hip fracture				
Non-COPD patients	3,170	2.26 (2.18 – 2.34)	Reference	Reference
Patients with COPD	1,213	3.57 (3.38 – 3.78)	1.67 (1.56 – 1.80)	1.09 (0.98 – 1.21) ^b

HR – Hazard ratio; CI – Confidence interval

HR – conditional regression used to account for matching by age, sex and GP practice.

Fully adjusted:

^a Multivariable Cox regression model derived HR was adjusted for age, sex, GP practice, Charlson Comorbidity Index, Body Mass Index, smoking status, inhaled corticosteroid use, antidepressant use and cumulative oral corticosteroid use.

^b Multivariable Cox regression model derived HR was adjusted for age, sex, GP practice, Charlson Comorbidity Index, Body Mass Index, smoking status, inhaled corticosteroid use and cumulative oral corticosteroid use.

**Table 3: Discrimination measures for FRAX® and QFracture® at recommended**
**treatment cut offs for both major osteoporotic fractures (MOF) and hip fractures alone**

Discriminatory measures	FRAX®	QFracture®
	Measure for ≥ 20% risk (95% CI)	Measure for ≥ 20% risk (95% CI)
All major Osteoporotic fractures (MOF)		
Sensitivity	25.4% (22.7-28.1%)	25.2% (22.5-27.9%)
Specificity	92.6% (91.0-94.2%)	87.7% (85.7-89.7%)
Positive Predictive Value	18.8% (16.4-21.1%)	12.2% (10.2-14.2%)
Negative Predictive Value	94.8% (93.4-96.2%)	94.5% (93.1-95.9%)
	Measure for ≥ 3% risk	Measure for ≥ 3% risk
Hip fracture		
Sensitivity	78.1% (75.6-80.7%)	82.1% (79.7-84.5%)
Specificity	60.8% (57.8-63.8%)	55.2% (52.1-58.3%)
Positive Predictive Value	3.9% (2.7-5.1%)	3.6% (2.5-4.8%)
Negative Predictive Value	99.3% (98.8-99.8%)	99.3% (98.8-99.8%)

CI – Confidence interval

**Figure 1: Kaplan-Meier plots comparing incidence of major osteoporotic**
**fractures (MOF) at various predicted fracture risk categories in patients**
**with COPD using (a) FRAX® and (b) QFracture®**

For peer review only

(a)

(b)

Appendix 1

Read code definitions for selected input variables

Variable	Read codes
COPD	H3...00, H3...11, H31..00, H310.00, H310000, H310z00, H311.00, H311000, H311100, H311z00, H312.00, H312000, H312011, H312100, H312300, H312z00, H313.00, H31y.00, H31y100, H31yz00, H31z.00, H32..00, H320.00, H320000, H320100, H320200, H320300, H320311, H320z00, H321.00, H322.00, H32y.00, H32y000, H32y100, H32y111, H32y200, H32yz00, H32z.00, H36..00, H37..00, H38..00, H39..00, H3A..00, H3y..00, H3y..11, H3z..00, H3z..11
Osteoporosis	5850.00, 58E4.00, 58E8.00, 58EA.00, 58EE.00, 58EG.00, 58EK.00, 58EM.00, 58ES.00, 58EV.00, 7230A, 7230B, 7230D, 7230PM, 7230PT, N330.00, N330000, N330100, N330200, N330300, N330400, N330500, N330600, N330700, N330800, N330900, N330A00, N330B00, N330C00, N330D00, N330z00, N331200, N331300, N331400, N331500, N331600, N331800, N331900, N331A00, N331B00, N331M00, N331N00, NyuB000, NyuB100, NyuB200, NyuB800
Antiresorptive treatment (drug code)	97138998, 99158998, 99158997, 97139998, 96920998, 96789998, 93478998, 97140998, 97218998, 93975992, 83457998, 97064992, 83456998, 96897998, 96020992, 96901998, 95879992, 98249990, 97031992, 98581990, 99018990, 98198990, 62945979, 96737998, 97066992, 97051992, 97780990, 98199990, 61594979, 99261990, 96604992, 92004979, 97248990, 99263990, 94089992, 93127992, 94756992, 91526998, 89828998, 88144998, 88144997, 88225998, 89434998, 93502998, 99862998, 95304998, 93228997, 96904998, 93228998, 95304996, 99862997, 95304997, 93228996, 99864998, 91997998, 91998998, 87933998, 81073998, 61612979, 87155998, 87154998, 88542998, 91378998, 82066998,

Antiresorptive treatment (drug code)	82065998, 81256998, 81255998, 91190996, 89518998, 91190998, 91191998, 86599998, 91190997, 91191997, 93692990, 81472998, 94276990, 93827990, 92431990, 94161990, 93610990, 94245990, 61524979, 99883979, 93828990, 99867979, 95572998, 99758998, 96764998, 97398992, 95244990, 89367998, 86562998, 86561998, 87645998, 87644998, 86079998, 86076998, 91533998, 87151998, 81270998, 91027998, 93617996, 93618996, 93618997, 93617997, 90527998, 86566998, 91028998, 87137998, 87136998, 91674998, 86564998, 86567998, 87135998, 93089979, 99357998, 84212998, 84691998, 89021998, 91764998, 90551998, 91763998, 81869998, 91764997, 91763997, 89354979, 92813997, 93402998, 92813998, 98527996, 93403996, 98527998, 93403998, 93402996, 84531998, 58602979, 87606998, 85936998, 81112998, 97865998, 85935998, 81111998, 76983978, 83078978
--	--

Appendix 2

METHODS

Potential confounders

For smoking status, alcohol use, MRC Dyspnoea scale, and a list of prescription drugs, the most recent record within 1 year (before and after) of index date were used. A BMI record within 2 years (before and after) of index date was used.

Where possible BMI was calculated from height and weight records, for patients with a missing BMI record. The BMI was subsequently categorised (underweight: <18.5 kg/m², normal: 18.5 - <25 kg/m², overweight: 25 - <30 kg/m², obese: >30 kg/m²).

Having received at least one prescription for inhaled corticosteroids, anti-epileptics, antidepressants, oestrogen-only Hormone Replacement Therapy (HRT) and osteoporosis medications, within 1 year (before and after) of index date were considered as risk factors.

Prediction tools – Input variables

The respective variable definitions as outlined in the algorithms for the prediction tools were used.

Smoking status – In QFracture[®], three current smoking categories are provided according to the number of cigarettes smoked daily[1]. To avoid the bias of categorising patients in one of the outlying categories, “current smokers” with no documented number of cigarettes smoked were assigned to the middle category “10-19 cigarettes daily” as done in a recent publication [2]. For FRAX[®]'s two-category smoking status, former smokers were assigned to the “non-smoker” category as was done in the cohorts used to develop FRAX[®]. [3]

Alcohol consumption – similarly, for alcohol use in QFracture[®], alcohol drinkers with no documented unit/day intake were assigned to “moderate (3-6units/day)”.

Missing values for BMI, smoking status, and alcohol use were imputed by multiple imputation using all predictors, resulting in twenty imputed datasets.[4] A complete case sensitivity analysis without imputed variables was also performed (Appendix 3).

References

- 1 ClinRisk Ltd. QFracture-2016® risk calculator. <http://www.qfracture.org/> (accessed 20 Sep 2017).
- 2 Dagan N, Cohen-Stavi C, Leventer-Roberts M, *et al.* External validation and comparison of three prediction tools for risk of osteoporotic fractures using data from population based electronic health records: retrospective cohort study. *BMJ* 2017;**356**:i6755. doi:10.1136/BMJ.I6755
- 3 Kanis JA, Oden A, Johnell O, *et al.* The use of clinical risk factors enhances the performance of BMD in the prediction of hip and osteoporotic fractures in men and women. *Osteoporos Int* 2007;**18**:1033–46. doi:10.1007/s00198-007-0343-y
- 4 Horton NJ, Lipsitz SR. Multiple Imputation in Practice. *Am Stat* 2001;**55**:244–54. doi:10.1198/000313001317098266

Appendix 3

Fracture risk prediction tools in COPD (Complete case analysis)

Of the 72,559 patients aged 40-90 years with COPD and no prior diagnosis of osteoporosis or prescription of any anti-resorptive treatment, 41,879 (57.7%) of patients had complete data. Amongst the patients with complete data, 2,649 (6.3%) experienced a MOF and 806 (1.9%) experienced hip fracture.

Both risk tools had about the same discriminatory accuracy as that obtained from the entire cohort with imputed data. The AUC for hip fracture was 75.6%, 95% CI 74.0% to 77.1% for FRAX® and 75.6%, 95% CI 74.0% to 77.2% for QFracture®. FRAX® maintained a higher accuracy for MOF (71.6%, 95% CI 70.6% to 72.6%) than QFracture® (61.1%, 95% CI 60.0% to 62.2%).

Appendix 4

Figure E1: Study population flow diagram

Appendix 5

Table E1: Risk of osteoporosis in patients with COPD compared with non-COPD patients

Descriptor	HR (95% CI)	Fully adjusted HR (95% CI)
COPD		
Non-COPD subjects	Reference	Reference
COPD patients	1.96 (1.87 – 2.06)	1.13 (1.05 – 1.22)
Charlson Comorbidity Index		
Score 0	Reference	Reference
Score 1	1.27 (1.18 – 1.36)	1.14 (1.06 – 1.23)
Score 2	1.34 (1.24 – 1.44)	1.27 (1.17 – 1.37)
Score 3	1.41 (1.28 – 1.55)	1.29 (1.17 – 1.42)
Score 4 & more	1.48 (1.33 – 1.64)	1.44 (1.29 – 1.61)
Body Mass Index (kg/m²)		
Underweight (<18.5)	1.93 (1.64 – 2.27)	1.91 (1.63 – 2.25)
Normal (18.5 – 24.9)	Reference	Reference
Overweight (25 – 29.9)	0.64 (0.60 – 0.69)	0.63 (0.58 – 0.67)
Obese (≥ 30)	0.47 (0.43 – 0.51)	0.45 (0.41 – 0.48)
No record	0.50 (0.46 – 0.53)	0.57 (0.52 – 0.61)
Smoking status		
Never	Reference	Reference
Ex	1.01 (0.95 – 1.08)	1.02 (0.95 – 1.09)
Current	1.23 (1.13 – 1.33)	1.15 (1.06 – 1.25)
Unknown	0.69 (0.64 – 0.74)	0.77 (0.71 – 0.83)
Oral Corticosteroid Use		
Unexposed	Reference	Reference
Exposed	2.79 (2.56 – 3.05)	1.91 (1.73 – 2.10)
Inhaled Corticosteroid Use		
No	Reference	Reference
Yes	1.35 (1.26 – 1.45)	1.24 (1.15 – 1.34)

HR – Hazard ratio; CI – Confidence interval

HR – Cox regression model derived HR adjusted for age, sex, and GP practice

The fully adjusted Hazard Ratio (aHR) was 1.13, 95% CI 1.05 to 1.22, $p < 0.0001$ – the multivariable Cox regression model derived aHR was adjusted for age, sex, GP practice, Charlson comorbidity index, body mass index, smoking status, inhaled corticosteroid use, and cumulative oral corticosteroid use.

Appendix 6

Table E2: Baseline characteristics of patients with COPD aged 40-90 years with no prior diagnosis of osteoporosis or prescription of any anti-resorptive treatment

Descriptor	COPD patients	
	n = 72,559	%
Mean age at index date (years, SD)	66.1 (10.7)	
Sex		
Female	31,885	43.9
MRC Dyspnoea Scale (1 Year either side of diagnosis)		
1	8,882	12.2
2	17,718	24.4
3	9,257	12.8
4 & 5	4,346	6.0
No record	32,356	44.6
Charlson Comorbidity Index Score		
0	0	0
1	38,573	53.2
2	11,953	16.5
3	11,110	15.3
≥ 4	10,923	15.1
Body Mass Index (BMI) (kg/m²)		
Underweight (< 18.5)	2,730	3.8
Normal (18.5 – 24.9)	21,791	30.0
Overweight (25 – 29.9)	21,504	29.6
Obese (≥30)	17,627	24.3
No BMI	8,907	12.3
Smoking status (1 Year either side of diagnosis)		
Never smoked	7,062	9.7
Ex-smoker	33,810	46.6
Current smoker	29,949	41.3
Unknown	1,738	2.4

STROBE 2007 (v4) Statement—Checklist of items that should be included in reports of *cohort studies*

Section/Topic	Item #	Recommendation	Reported on page #
Title and abstract	1	(a) Indicate the study's design with a commonly used term in the title or the abstract	1
		(b) Provide in the abstract an informative and balanced summary of what was done and what was found	2
Introduction			
Background/rationale	2	Explain the scientific background and rationale for the investigation being reported	5
Objectives	3	State specific objectives, including any pre-specified hypotheses	5
Methods			
Study design	4	Present key elements of study design early in the paper	6
Setting	5	Describe the setting, locations, and relevant dates, including periods of recruitment, exposure, follow-up, and data collection	6
Participants	6	(a) Give the eligibility criteria, and the sources and methods of selection of participants. Describe methods of follow-up	6
		(b) For matched studies, give matching criteria and number of exposed and unexposed	6
Variables	7	Clearly define all outcomes, exposures, predictors, potential confounders, and effect modifiers. Give diagnostic criteria, if applicable	6,7
Data sources/ measurement	8*	For each variable of interest, give sources of data and details of methods of assessment (measurement). Describe comparability of assessment methods if there is more than one group	6,7
Bias	9	Describe any efforts to address potential sources of bias	7
Study size	10	Explain how the study size was arrived at	-
Quantitative variables	11	Explain how quantitative variables were handled in the analyses. If applicable, describe which groupings were chosen and why	7
Statistical methods	12	(a) Describe all statistical methods, including those used to control for confounding	7
		(b) Describe any methods used to examine subgroups and interactions	-
		(c) Explain how missing data were addressed	6, Appendix 2
		(d) If applicable, explain how loss to follow-up was addressed	-
		(e) Describe any sensitivity analyses	7
Results			
Participants	13*	(a) Report numbers of individuals at each stage of study—eg numbers potentially eligible, examined for eligibility, confirmed	9

		eligible, included in the study, completing follow-up, and analysed	
		(b) Give reasons for non-participation at each stage	-
		(c) Consider use of a flow diagram	-
Descriptive data	14*	(a) Give characteristics of study participants (eg demographic, clinical, social) and information on exposures and potential confounders	9, Table 1 (17)
		(b) Indicate number of participants with missing data for each variable of interest	Table 1 (19)
		(c) Summarise follow-up time (eg, average and total amount)	9
Outcome data	15*	Report numbers of outcome events or summary measures over time	9, 10
Main results	16	(a) Give unadjusted estimates and, if applicable, confounder-adjusted estimates and their precision (eg, 95% confidence interval). Make clear which confounders were adjusted for and why they were included	9, 10, Table 2 (20), Appendix 5
		(b) Report category boundaries when continuous variables were categorized	Appendix 2
		(c) If relevant, consider translating estimates of relative risk into absolute risk for a meaningful time period	-
Other analyses	17	Report other analyses done—eg analyses of subgroups and interactions, and sensitivity analyses	9,10,11
Discussion			
Key results	18	Summarise key results with reference to study objectives	12
Limitations			
Interpretation	20	Give a cautious overall interpretation of results considering objectives, limitations, multiplicity of analyses, results from similar studies, and other relevant evidence	12-14
Generalisability	21	Discuss the generalisability (external validity) of the study results	14
Other information			
Funding	22	Give the source of funding and the role of the funders for the present study and, if applicable, for the original study on which the present article is based	15

*Give information separately for cases and controls in case-control studies and, if applicable, for exposed and unexposed groups in cohort and cross-sectional studies.

Note: An Explanation and Elaboration article discusses each checklist item and gives methodological background and published examples of transparent reporting. The STROBE checklist is best used in conjunction with this article (freely available on the Web sites of PLoS Medicine at <http://www.plosmedicine.org/>, Annals of Internal Medicine at <http://www.annals.org/>, and Epidemiology at <http://www.epidem.com/>). Information on the STROBE Initiative is available at www.strobe-statement.org.